

# Decoding pelagic ciliate (Protozoa, Ciliophora) community divergences in size spectrum, biodiversity and driving factors spanning global five temperature zones

Chaofeng Wang[1,2,3,4], Zhiqiang Xu[1,5], Guangfu Luo[6], Xiaoyu Wang[7], Yan He[8], Musheng Lan[6],
Tiancheng Zhang[2], Wuchang Zhang[1,3,4]

[1]CAS Key Laboratory of Marine Ecology and Environmental Sciences, Institute of Oceanology, Chinese Academy of Sciences, Qingdao, 266071, China
[2]State Key Laboratory of Mariculture Breeding, Key Laboratory of Marine Biotechnology of Fujian Province, Institute of Oceanology, College of Marine Sciences, Fujian Agriculture and Forestry University, Fuzhou, 350002, China
[3]Laboratory for Marine Ecology and Environmental Science, Qingdao National Laboratory for Marine Science and Technology, Qingdao, 266237, China
[4]Center for Ocean Mega-Science, Chinese Academy of Sciences, Qingdao, 266071, China
[5]Jiaozhou Bay Marine Ecosystem Research Station, Institute of Oceanology, Chinese Academy of Sciences, Qingdao, 266071, China
[6]Polar Research Institute of China, Shanghai, 200136, China
[7]Frontiers Science Center for Deep Ocean Multispheres and Earth System, Key Laboratory of Physical Oceanography, Ocean University of China, Qingdao 266100, China
[8]First Institute of Oceanography and Key Laboratory of Marine Science and Numerical Modeling, Ministry of Natural Resources, Qingdao, 266061, China

*Correspondence to*: Chaofeng Wang (wangchaofeng@qdio.ac.cn); Wuchang Zhang (wuchangzhang@qdio.ac.cn)

**Abstract.** Community structure of microzooplanktonic ciliate in size spectrum, biodiversity and biotic-abiotic interplay are essential components for unraveling their ecological role in marine ecosystems, yet remain challenging to elucidate on a global scale. To address this knowledge gap, we conducted field observational studies across five temperature zones (North Frigid Zone, NFZ; Sub-Arctic Zone, SAZ; North Temperate Zone, NTZ; Torrid Zone, TZ; South Frigid Zone, SFZ). Our
analysis revealed that a sharply decline in ciliate abundance and biomass occurred at 100 m layer, with distinct vertical distribution patterns observed in each climate region. Moreover, ciliate size spectra exhibited a decrease trend from small to large size spectra, with steeper slopes observed in the NFZ and SFZ compared to the other temperature zones. Furthermore, an anti-phase relationship between ciliate abundance and tintinnid biodiversity was observed in latitudinal direction, with the TZ and bipolar seas characterized by the highest biodiversity and abundance, respectively. Moreover, a multivariate biota-
environment analysis indicated that temperature exert a primary influence on microzooplanktonic ciliates in the global marine ecosystem, and the bottom-up control play a key role in shaping ciliate community. In conclusion, these results underscore the unprecedented divergences in ciliate trait structure among five temperature zones and can be generalised for assessing the potential effects of climate change on pelagic microzooplankton in future marine realm.



## 1 Introduction

The Earth is traditionally divided into five temperature zones based on established climate classifications: the North Frigid Zone (NFZ), North Temperate Zone (NTZ), Torrid Zone (TZ), South Temperate Zone (STZ), and South Frigid Zone (SFZ) (Köppen 1936; Trewartha et al. 1967). Therein, each temperature zone possessed unique ocean circulation pattern and concurrent specific plankton biome (Longhurst 2007; Spalding et al. 2012). Albeit a myriad of prevailing research relevant to plankton biogeography and its interplay with environmental drivers highlighting its importance in disentangling marine

ecosystems and biogeochemical cycles (e.g., Wang et al. 2020; Darnis et al. 2022; Segaran et al. 2023; Tagliabue et al. 2023), substantial global-scale studies were conducted through diverse modeling approaches (Spalding et al. 2012; Blanchard et al. 2017; Anderson et al. 2021; Benedetti et al. 2021; Heneghan et al. 2023; Atkinson et al. 2024). To date, an explicit and comprehensive representation of plankton community trait structure using data-derived statistical analysis originated from field-surveys remains unresolved.

A holistic paradigm of plankton biogeography across marine ecosystem is crucial for deciphering global ecological connectivity (Hillman et al. 2018) and predicting how ecosystems respond to stressors induced by climate change (Darnis et al. 2022). Over recent decades, anthropogenic CO2 emissions have led to increased atmospheric concentrations and greater global radiative forcing (Tagliabue et al. 2023), triggering diverse ecological feedbacks worldwide, for instance poleward distribution shifts (Neukermans et al. 2018; Oziel et al. 2020; Benedetti et al. 2021), adjustments in phenology (Poloczanska

et al. 2013; Atkinson et al. 2015; Chust et al. 2024), and reductions in mean body size (Daufresne et al. 2009; Verberk et al. 2021; Wang et al. 2023a, 2023b). In this sense, extensive existing studies put emphasis on biotic community response to climate change in the bipolar and adjacent seas owing to their higher susceptibility compared to tropical, subtropical, and temperate seas (Serreze et al. 2009; Screen and Simmonds 2010; IPCC 2023; Noh et al. 2024). Unfortunately, an informative research relate to environmental affinity of plankton, particularly microzooplankton, is not sufficiently understood in

aforementioned five temperature zones.

In the realm of microzooplankton, pelagic ciliates stand out as the predominant biological entities, spanning in size from 10 to 200 μm, and hold significant sway over both biodiversity and abundance, particularly in the polar and adjacent seas (Taniguchi 1984; Strom and Fredrickson 2008; Lu and Weisse 2022; Kohlbach et al. 2023; Wang et al. 2023a, 2024a, 2024b). Taxonomically categorized within the phylum Ciliophora, class Spirotrichea, and subclasses Oligotrichia and

Choreotrichia, pelagic ciliates, including aloricate ciliates and tintinnids, are ubiquitous single-cell protozoans found in various aquatic environments worldwide (Lynn 2008). Furthermore, ciliates play an irreplaceable role in marine trophodynamics (carbon cycle and energy transfer) through prey-predator interactions, serving as both phytoplankton grazers and prey for metazoans (Stoecker et al. 1987; Dolan et al. 1999; Calbet and Saiz 2005; Gómez 2007; Weisse and Sonntag 2016). Specifically, owing to their simple life cycle, fast-reaction to environmental changes, and strong adaptability, pelagic

ciliates, particularly tintinnids, are widely recognized as ideal bioindicators for assessing various sea conditions (e.g., Kato and Taniguchi 1993; Jiang et al. 2013; Wang et al. 2021; Yu et al. 2022).





Recent escalation in global warming have imposed a cascade of impacts on aquatic ecosystems, presenting a formidable challenge to inherent holopelagic species that project the relevant adaptative strategies (Stabeno et al. 2012; Yasumiishi et al. 2020; Carvalho et al. 2021; Atkinson et al. 2024). Accordingly, a prevailing viewpoint for phytoplankton, the cornerstone of

marine pelagic food web, is a major decline in both biomass and size spectra in the NTZ, TZ and STZ (Li et al. 2009; Lotze et al. 2019; Tittensor et al. 2021), leading to subsequent declines for higher trophic levels, termed "trophic amplification" (Kwiatkowski et al. 2019; du Pontavice et al. 2021). As grazer of pelagic phytoplankton, response of microzooplanktonic ciliate to ocean warming in the bipolar and adjacent seas is substantial (Li et al. 2022; Wang et al. 2022a, 2023a, 2023b, 2024b), yet comparative assessments amid their trait structure (e.g., size spectra, biodiversity and biotic-abiotic interplay)

remain unexplored to date.

Hence, focusing on microzooplanktonic ciliate size spectra, species diversity and biotic-abiotic interplay in global-scale for future marine ecosystem dynamic projections could bolster our understanding of plankton response to sophisticated climate changes, particularly as the underlying microbial processes remain poorly resolved. Here, we propose a hypothesis that variations in hydrographic conditions are likely responsible for the diverse ciliate trait structures observed globally. By

optimizing feild observational data and available methods, our objective is two-fold: disclose the microzooplanktonic ciliate adaptative strategies to alien hydrography among temperature zones; and evaluate the potential dynamics of microzooplankton to rapid climate change. Given the current foreseeable rapid climate change process, this study will offer a valuable norm for facilitating the phenological and bioclimatic progression of microzooplankton in future global marine ecosystem realm.

## 2 Materials and Methods

### 2.1 Study area and field sampling

Based on their latitudinal locations, field samplings of microzooplanktonic ciliate were conducted in five temperature zones (Trewartha et al. 1967): 1, North Frigid Zone (NFZ), encompassing the Arctic Ocean, during July to August 2019 and 2023 aboard the *R.V.* "Xiangyanghong 01" and *R.V.* "Xuelong 2", respectively; 2, the Sub-Arctic Zone (SAZ), located in the

Bering Sea, in July to August 2019 aboard the *R.V.* "Xiangyanghong 01"; 3, the North Temperate Zone (NTZ), situated in the North Pacific, in September 2019 aboard the *R.V.* "Dongfanghong 3"; 4, the Torrid Zone (TZ), which includes the tropical western Pacific in December 2016 and August 2017 aboard the *R.V.* "Kexue", and the Indian Ocean in March 2021 aboard the *R.V.* "Xiangyanghong 10"; and 5, the South Frigid Zone (SFZ), covering the Southern Ocean, from December 2020 to March 2021 aboard the *R.V.* "Xuelong 2" (Figure 1). A total of 1117 samples (175 stations along 19 transects) were

sampled.

Seawater samples were collected with a rosette sampler carrying 24 Niskin bottles (each 12 L). Simultaneously, environmental factors of sampling depth, temperature, salinity and chlorophyll *a in vivo* fluorescence (Chl *a*) were obtained by a multi-sensor profiler (CTD–SeaBird SBE 911) at each cruise. All microzooplanktonic ciliate samples (except the SAZ,





where seafloor of most stations were shallower than 200 m) were collected at surface (2 m), 25 m, 50 m, 75 m, 100 m, 150 m

and 200 m at each designated station. Furthermore, each sample was fixed with acid Lugol's (1% final concentration) and preserved in darkness at 4 ℃ until further analysis in the laboratory.

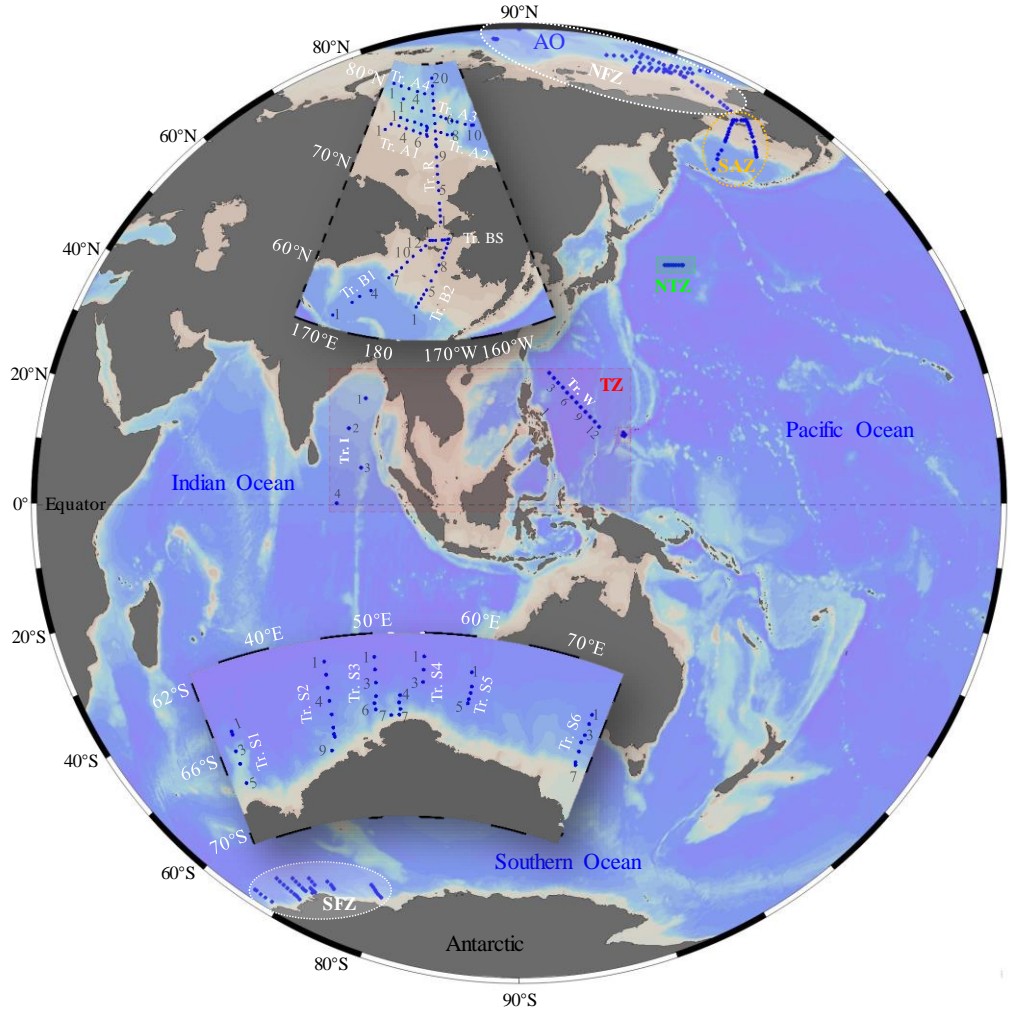

**Figure 1: Survey stations and transects (Tr.) in the tropical, temperate and bipolar seas. AO, Arctic Ocean; NFZ, North Frigid Zone; SAZ, Sub-Arctic Zone; NTZ, North Temperate Zone; TZ, Torrid Zone; SFZ, South Frigid Zone.**

**2.2 Sample analysis**

In the laboratory, each sample was concentrated to approximately 200 mL by siphoning off supernatant after settling 60 h. After two rounds of siphon process, a final of 25 mL highly concentrated sample was obtained, and then settled in a Utermöhl counting chamber (Utermöhl 1958). Each sample in the chamber was examined using an Olympus IX 71 inverted microscope (100× or 400×), and total abundance, body-size and species richness of ciliates (including aloricate ciliates and



tintinnids) were recorded at five temperature zones by Chaofeng Wang. To ensure accuracy, cellular size (e.g., length, width, shape) of aloricate ciliate or each tintinnid species were measured for at least 10 individuals if possible.

Additionally, body-size of both aloricate ciliates and tintinnids were categorized into 10 μm increments (10–20 μm, 20–30 μm, etc.) based on body length (Wang et al., 2020), and further classified into small (10–20 μm)/medium (20–50 μm)/large (>50 μm) size-fractions following Yang et al. (2019). Regarding species richness, tintinnid identification was assigned to

closest species as described in Zhang et al. (2012). Furthermore, we select the average value (15, 25, 35, 45 μm,…, etc) of each size-fraction of both loricate ciliate and tintinnid as the counting criterion for ciliate size spectra (Wang et al. 2024b).

**2.3 Data processing**

Ciliate volumes were estimated according to their appropriate geometric shapes (cone, ball, cylinder). Carbon biomass of each tintinnid was calculated by the equation (Verity and Lagdon 1984):

$C = V_i \times 0.053 + 444.5$

Where $C$ ($10^{-6}$ μg C) was the carbon biomass of individual tintinnid, $V_i$ (μm$^3$) was the lorica volume. Additionally, a conversion factor (0.19 pg C μm$^{-3}$) was used for calculating aloricate ciliate carbon biomass (Putt and Stoecker 1989). Size spectra of both ciliate abundance and biomass were classified based cellular length. Furthermore, in order to better unravelling test tintinnid biodiversity spanning five temperature zones, the Margalef index ($d_{Ma}$) (Margalef 1958) (1) and

Shannon index ($H_2'$) (Shannon 1948) (2) were conducted by the following equations:

$d_{Ma} = \frac{S-1}{\ln N}$ (1)

where $S$ is the number of species, and $N$ is the total number of tintinnid individuals in the sample.

$H_2' = -\sum_{i-1}^{S} P_i \log_2 P_i$ (2)

where $S$ is the number of species, and $N$ is the total abundance of tintinnid individuals in the sample. $P_i$ ($N_i/N$) is the relative

abudance of $i$ species in a whole community.

Biogeographically, classification of tintinnid genera (cosmopolitan, warm water, boreal, austral and neritic) was based on Pierce and Turner (1993) and Dolan and Pierce (2013). Among them, tintinnid genera were further classified into oceanic (cosmopolitan, warm water, boreal and austral) and neritic types. Moreover, average value of each parameter was represented as mean ±SD in the following text.

Hereinafter, sampling map was visualized by ODV (Ocean Data View, Version 4.7), and ciliate distributional data of size–diversity and temperature – diversity relationships were analyzed using Surfer (Version 13.0), Grapher (Version 12.0), and OriginPro 2021 (Version 9.6). Moreover, the Biota-Environment analysis was performed based on Spearman's correlation between log-transformed abiotic parameters and square root-transformed abundance data (t-test) using both PRIMER (Version 5.0) and OriginPro 2021 (Version 9.6). Additionally, the slope of the size spectrum (a straight line fitted through

the size spectrum on a log–log plot) (Blanchard et al. 2017) was carried out to quantize its interplay with ciliate abundance at discrete depth of aforementioned global seas (95% confidence). In the following, based on the slope condition, we use the





decreasing rate ($\Delta_D$) or increasing rate ($\Delta_I$) according to ciliate abundance or species richness and environmental variables to quantize their interplay in the global seas.

## 3 Results

**3.1 Hydrography and ciliate abundance and biomass**

Each environmental parameter (temperature, salinity, and Chl *a*) displayed distinct spatiotemporal variations globally (Figure 2 and Figures S1-S3). Horizontally, at surface, 50 and 100 m layers, both temperature and salinity peaked in the Torrid Zone (TZ), contrasting with Chl *a*, which exhibited its lowest value in the same region (Figure 2 and Figures S1-S2). At 200 m depth, temperature and Chl *a* peaked in the TZ and North Frigid Zone (NFZ), respectively, deviating from salinity

patterns, which exhibited high values in both the TZ and NFZ (Figure 2 and Figure S1). Vertically, temperature and Chl *a* declined in the NFZ and Sub-Arctic Zone (SAZ), while salinity increased from the surface to 200 m layers across all regions (Figures S1-S3). Moreover, temperature displayed a "sandwich" structure (low–high–low values) at inner stations of the South Frigid Zone (SFZ), and Chl *a* peaked at subsurface layers in both the North Temperate Zone (NTZ) and TZ (Figures S1 and S3).

Pelagic ciliate abundance ranged from 22–9142 ind. L$^{-1}$ in the NFZ, 182–9242 ind. L$^{-1}$ in the SAZ, 65–886 ind. L$^{-1}$ in the NTZ, 25–436 ind. L$^{-1}$ in the TZ, and 44–5866 ind. L$^{-1}$ in the SFZ, whereas their biomass ranged from 0.0–39.3 µg C L$^{-1}$, 0.3–24.0 µg C L$^{-1}$, 0.1–1.1 µg C L$^{-1}$, 0.0–1.1 µg C L$^{-1}$, and 0.0–26.1 µg C L$^{-1}$ in aforementioned regions, respectively (Figure 2 and Figures S1-S3). Horizontally, both high abundance ($\geq$ 2000 ind. L$^{-1}$) and biomass ($\geq$ 5.0 µg C L$^{-1}$) of ciliates were observed in surface layers of the NFZ, SAZ, and SFZ, coinciding with high Chl *a* levels. At 50 m, 100 m and 200 m layers,

the SAZ and TZ had the highest and lowest abundance, respectively (Figure 2 and Figure S1). Vertically, both ciliate abundance and biomass exhibited a surface-peak pattern in the NFZ, SAZ, and SFZ, whereas in the NTZ and TZ, this pattern transitioned to subsurface-peak and bimodal-peak distributions, respectively (Figures S1 and S3).

Meanwhile, aloricate ciliates dominated the ciliate community, accounting for $\geq$ 90% of total abundance at each depth in the NFZ, NTZ, TZ, and SFZ. However, in the SAZ, tintinnid played a more significant role in the ciliate community, with an

average relative abundance at most sampling depths exceeding 10% (Figures S4). In terms of aloricate ciliates in the horizontal direction, small (10–20 µm) and medium (20–50 µm) size-fractions in the SAZ exhibited the highest average abundance at surface, 50 m, 100 m, and 200 m layers, whilst the largest (> 50 µm) size-fraction had the highest average abundance at the surface, 50 m, and 100 m layers in the SFZ (Figures S5). Additionally, except for the NTZ, the abundance and relative abundance of the medium size-fraction were highest in the other four regions at both the surface and 50 m layers.

At 200 m depth, the small size-fraction predominated among the aloricate ciliates (Figures S5). Vertically, the large (> 50 µm) and small size-fractions exhibited an inverse distribution characteristic across five temperature zones (Figures S5).



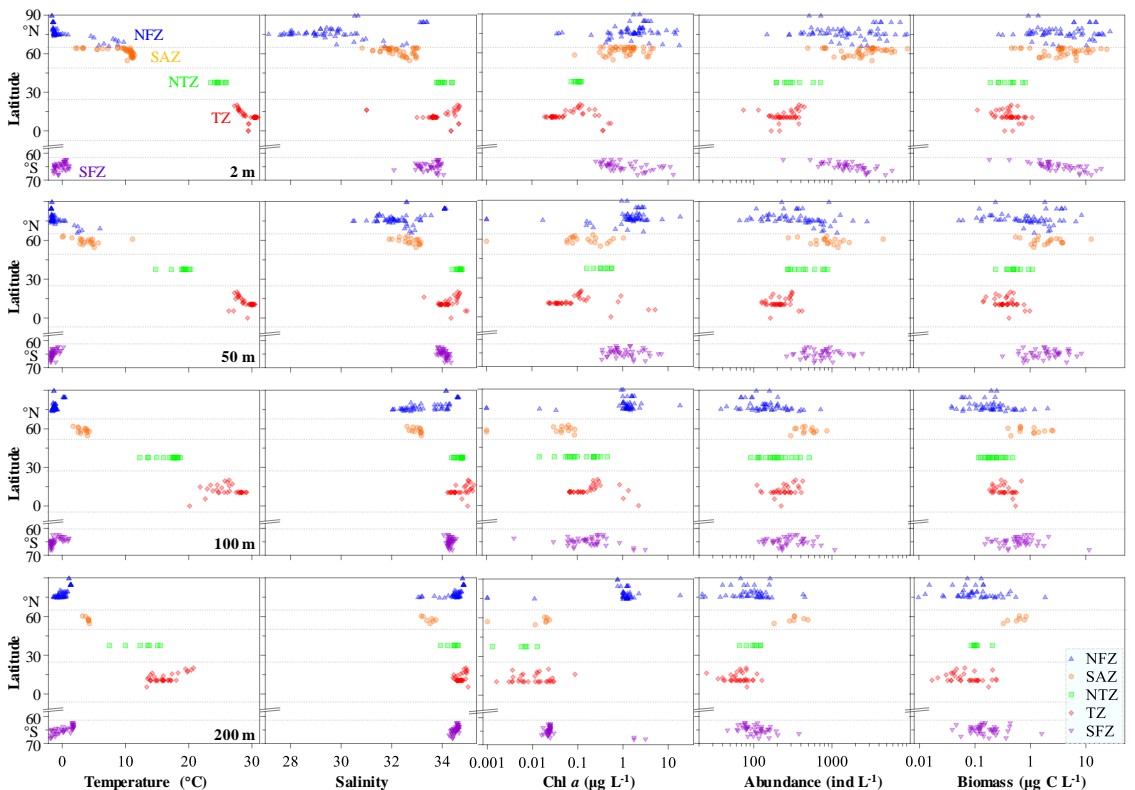

**Figure 2: Variations in environmental variables and ciliate abundance and biomass at discrete depth in the North Frigid Zone (NFZ), sub-Arctic Zone (SAZ), North Temperate Zone (NTZ), Torrid Zone (TZ) and South Frigid Zone (SFZ).**

### 3.2 Notable variations in pelagic ciliate size spectrum composition

The abundance and biomass of pelagic ciliate size spectra displayed significant variations across global seas (95% confidence) (Figure 3). Generally, the slopes of the normalized abundance and biomass size spectra varied from -2.13 to -0.87 (average -1.60±0.33), and from -0.99 to -0.08 (average -0.53±0.25), respectively, with the former was much steeper than tha latter (Figure 3). Therein, ciliate abundance decreased from small (15 μm) to large size spectra (> 100 μm), with the slopes of the normalized abundance size spectra in both the NFZ (-2.13 to -1.93, average -2.01±0.09) and SFZ (-2.01 to -1.63, average -1.80±0.17) being steeper than in the other three regions at each depth (Figure 3a). Additionally, a secondary peak in abundance, featuring large size spectra (> 100 μm), was observed at the surface layers of the NFZ, SAZ, and SFZ (Figure 3a). In contrast, the distribution characteristics of ciliate biomass within size spectra did not align with the abundance trend (Figure 3b). Notably, the 65 μm size spectrum exhibited the highest values at both surface and 50 m layers of the NFZ, followed by the SFZ (55 μm) and SAZ (55 μm), with the TZ (35 μm) and NTZ (25 μm) showing lower values (Figure 3b). Moreover, the slopes of the normalized biomass size spectra in the SFZ (-0.99 to -0.77, average -0.86±0.10) were steeper than that in the SAZ (-0.74 to -0.43, average -0.62±0.13), NTZ (-0.63 to -0.44, average -0.53±0.09), TZ (-0.74 to -0.25, average -0.47±0.22) and NFZ (-0.37 to -0.08, average -0.21±0.12) (Figure 3b). Interestingly, the highest biomass of ciliate





size spectra at the surface, 50 m, and 100 m layers of the TZ corresponded to the 35 μm size spectrum, while at the 200 m

layer, the 15 μm size spectrum became dominant (Figure 3b).

**Figure 3: Variations in body-size spectra of ciliate normalized abundance (a) and biomass (b) at discrete depth in each temperature zone.**



### 3.3 Dynamics in tintinnid species richness and diversity indices

Tintinnid species richness and diversity indices exhibited obvious variations in both horizontal and vertical distributions
spanning five temperature zones (Figure 4 and Figure S6). Horizontally, species richness, Margalef index ($d_{Ma}$) and Shannon
index ($H_2'$) were notably high at discrete layers in both the NTZ and TZ, followed by the SAZ, NFZ, and SFZ (Figure 4a and
Figure S6). Given that neritic genera were only present in the SAZ and NFZ, species richness excluding neritic genera was
also examined to maintain consistency with the other three regions, revealing higher species richness in the SFZ compared to

the NFZ (Figure 4a). Vertically, elevated values of tintinnid species richness, dMa and H2′ were primarily observed in the
upper 50 m waters of the NFZ, SAZ, and SFZ, while these values peaked at 75 m and 100 m in the NTZ and TZ,
respectively (95% confidence) (Figure 4b). Furthermore, an analysis of the relationship between ciliate abundance and
species richness revealed that regions characterized by high ciliate abundance were often accompanied by low tintinnid
species richness (Figure S7).

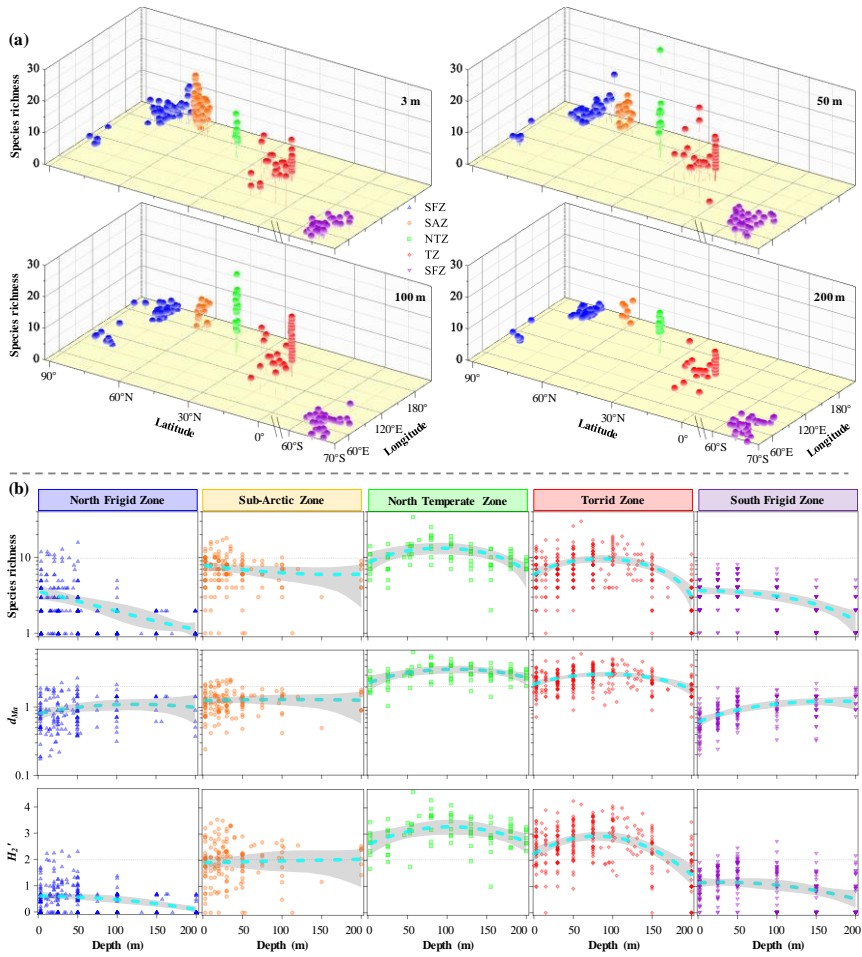


**Figure 4: Variations of tintinnid species richness, Margalef index ($d_{Ma}$) and Shannon index ($H_2'$) in latitudinal (a) and vertical (b) direction of all regions.**





## 3.4 Biotic-abiotic interplay and its variations

Ciliate abundance and tintinnid species richness exhibited varying correlations with environmental parameters across the five
temperature zones (Figures 5-7 and Figure S8). In terms of the biotic-abiotic interplay trend, our results revealed that only
the NFZ and SAZ exhibited an increasing trend ($\Delta_I \geq 0.03$) in abundance–temperature correlation at both surface and 50 m
layers compared to other three temperate zones (Figure 5a). Concerning all sampling layers, only the SFZ, differing from the
trends observed in the other four temperature zones, displayed a decrease in ciliate abundance with increasing temperature
($\Delta_D$ = -0.26, $R^2$ = 0.06) (Figure S8). Moreover, only the TZ and SFZ exhibited an increase ($\Delta_I \geq 0.29$) and a decrease ($\Delta_D \leq$ -
0.01) trend at each sampling layer in abundance–salinity correlation, respectively (Figure 5b). Furthermore, only SFZ
showed an increase ($\Delta_I \geq 0.02$) trend at each sampling layer in abundance–Chl $a$ correlation (Figure 5c), which was align
with trends in other four temperature zones at all sampling layers ($\Delta_I \geq 0.06$) (Figure S8).

Regarding species richness–temperature correlation, the highest increasing trend occurred at 50 m of the NFZ ($\Delta_I$ = 0.26, R2
= 0.44), while the highest decreasing trend was found at 100 m of the SAZ ($\Delta_D$ = -0.28, $R^2$ = 0.09) (Figure 6a). As for all
sampling layers, only the NFZ and TZ exhibited an increasing trend in species richness–temperature correlations, with the
former ($\Delta_I$ = 0.15, $R^2$ = 0.26) being higher than the latter ($\Delta_I$ = 0.06, $R^2$ = 0.23) (Figure S8). Moreover, concerning biotic–
salinity correlations, only the SAZ exhibited an increase ($\Delta_I \geq 0.06$) trend at each sampling layer (Figure 6b). At all sampling
layers, contrasting with decrease trend in abundance-salinity correlation in each temperature zone, the increase trends in
species richness-salinity correlations were observed in the NTZ ($\Delta_I$ = 0.45, $R^2$ = 0.08) and SAZ ($\Delta_I$ = 0.20, $R^2$ = 0.03) (Figure
S8). Furthermore, only the bipolar seas exhibited an increasing trend ($\Delta I \geq 0.01$) in species richness-Chl a correlation at each
sampling layer (Figure 6c). Simultaneously, species richness exhibited an increasing trend in their correlation with Chl a at
all sampling layers of the SAZ ($\Delta_I$ = 0.05, $R^2$ = 0.02), NTZ ($\Delta_I$ = 0.07, $R^2$ = 0.05) and TZ ($\Delta_I$ = 0.20, $R^2$ = 0.15) (Figure S8).

To further quantize the physical-biological interplay in five temperature zones, we conducted both principal component
analysis (PCA) and spearman's rank correlation via using abundance of aloricate ciliate, tintinnid and total ciliate, and
tintinnid species richness to test abiotic influence (Figure 7). The PCA revealed that two principal components effectively
differentiated the environmental conditions among five temperature zones. These components accounted for a substantial
proportion of the biotic variation in the NFZ (62.85%), SAZ (67.83%), NTZ (64.75%), TZ (72.68%), SFZ (63.84%) and all
regions (61.42%) (Figure 7a). Akin to PCA, spearman's rank correlation reflected that abundance of aloricate ciliate,
tintinnid and total ciliate in all five temperature zones displayed a strong significant negative and positive correlation with
depth ($p < 0.01$) and Chl $a$ ($p < 0.01$), respectively (Figure 7b). Furthermore, both aloricate ciliate and tintinnid featured
significant positive correlation with temperature in the SAZ, NTZ and TZ ($p < 0.05$). However, in the SFZ, relationship
between aloricate ciliate and temperature shifted to a significant negative correlation ($p < 0.05$) (Figure 7b). Except that,
tintinnid species richness exhibited strong significant negative correlation with salinity in both the NFZ and SFZ ($p < 0.01$),
which was unconsistent with that in the NTZ, where changed into strong significant positive correlation ($p < 0.01$) (Figure
7b).





Figure 5: Variations in slopes between ciliate abundance and environmental variables (temperature, salinity, Chl *a*) at discrete depth in each temperature zone.





Figure 6: Variations in slopes between tintinnid species richness and environmental variables (temperature, salinity, Chl *a*) at discrete depth in each temperature zone.





**Figure 7: Variations in principal component analysis (PCA) (a) and spearman's rank correlation (b) between environmental parameters (Depth; temperature, T; salinity, S; Chl *a*) and ciliate (tintinnid, Tin; aloricate ciliate, AC; total ciliate, TC; tintinnid species richness, SR) in five regions. The *x*-axis is the first PCA axis, and the *y*-axis is the second PCA axis. Environmental variables and ciliates are indicated by red lines and black lines, respectively. Grey dots are sampling points. $^{**}$: $p < 0.01$, $^{*}$: $p < 0.05$, t-test.**






## 4 Discussion

In a nutshell, this study provides a first holistic epitome of microzooplanktonic ciliate community divergences and
corresponding biotic-abiotic interplay among five temperature zones (NFZ, SAZ, NTZ, TZ, SFZ) spanning the global scale,
and decoded that the dynamics of microzooplanktonic ciliate trait structures were governed by unique physicochemical
feature in each temperature zone. Simultaneously, it is noteworthy that trait structures (including vertical distribution
patterns, latitudinal dynamics, size spectrum, and species diversity) of ciliates, analyzed through a ground-based data-driven
statistical approach, exhibited disproportionate variations among the five temperature zones mentioned (Figures 2-4).
Among these, abiotic parameters, particularly temperature, likely played a significant role in driving these variations, as
hypothesized (Chapin et al. 1997; Anderson et al. 2021; Tanioka et al. 2022; Jiao et al. 2024). Additionally, numerous
scientific cruises in China have provided sampling opportunities spanning a latitudinal gradient of biological "hotspot"
regions, which encompassing 175 sites in the NFZ, SAZ, NTZ, TZ and SFZ. However, it's essential to acknowledge that our
study areas cannot fully represent the diverse adaptative strategies of ciliates across these regions, particularly in the Atlantic
Ocean. Hence, further cruises and studies are necessary to conduct relevant experiments in the Atlantic Ocean in the near
future.

### 4.1 Significant divergences in functional trait of ciliate size spectrum

Plankton size spectrum, which represents the distribution of individuals in a community or ecosystem by numerical
abundance or biomass across size classes typically displayed on log axes, plays a crucial role in modulating various
microbial processes (e.g., carbon cycle driven by prey-predator interactions) (Garc á-Comas et al. 2016; Andersen 2019;
Trombetta et al. 2020; Serra-Pompei et al. 2022; Antoni et al. 2024; Atkinson et al. 2024), and can elucidate ecological
functions within marine food webs (Vandromme et al. 2012). In this sense, although empirical evidences give details on both
plankton size-spectra functional traits and concurrent valuable models, majority integrative analysis were more dedicated to
the biomass density size spectrum than their abundance amid different trophic levels (Sprules et al. 2016; Blanchard et al.
2017; Atkinson et al. 2024; Stukel et al. 2024). Currently, research on monospecific trophic group, such as
microzooplanktonic ciliates (Wang et al. 2024a), is rarely studied on a global scale. Similar to Stukel et al. (2024), our study
revealed that the slopes of abundance size spectra in both the NFZ and SFZ were steeper in bipolar seas than other three
regions latitudinally (Figure 3). Furthermore, the general trend of steeper slopes at the surface compared to the 200 m layer
across all regions suggests a community size shift influencing carbon flux efficiency towards higher trophic levels (Stukel et
al. 2024).

In addition, Stukel et al. (2024) depicted that the slopes of the normalized biomass size spectra varied from −1.6 to −1.2
(median slope was −1.4) spanning over five orders of magnitude from phytoplankton to macrozooplankton in plankton
communities in the tropical and subtropical seas. In contrast, our findings revealed the median slope was about −0.53 for the
biomass size spectrum (no clear straight line on a log-log plot) across all discrete depths of the global seas (Figure 3b). We



deem that the finer-scale monospecific trophic group, spanning one order of magnitude (10-200 μm, microzooplankton), might be too small to accurately calculate the slopes of the normalized biomass size spectra (Sheldon et al. 1972). Conversely, it's noteworthy that the slopes of the abundance size spectrum exhibited an inverse relationship between abundance and body-size (Figure 3a), resembling the pyramid of numbers concept (Elton 1927; Trebilco et al. 2013; Blanchard et al. 2017). Hence, we posit that the slope of the abundance size spectrum may be more informative than its

biomass counterpart in covering one order of magnitude within the plankton community. Moreover, the steeper slopes observed in the abundance size spectra in the bipolar seas compared to the tropical, temperate, and sub-Arctic seas might reflect a prevailing trend towards miniaturization (Li et al. 2009; Wang et al. 2023a).

### 4.2 Tintinnid biodiversity dynamics and its underlying formation mechanisms

By virtue of its critical role in regulating ecosystem processes and resource utilization efficiency, plankton species diversity

play a crucial role in marine ecosystem functioning and biogeochemical cycling (Chapin et al. 1997). Similarly, a higher functionally similar species diversity enhances stability in resistance and resilience aspects of marine ecosystem processes (Ibarbalz et al. 2019; Benedetti et al. 2021; Chust et al. 2024). Consistent with both observational and modeling studies, tintinnid biodiversity was highest in the tropical and subtropical seas, and was lowest in the bipolar seas (Figure 4) (e.g., Sherr et al. 1997; Dolan et al. 2013, 2014, 2016; Righetti et al. 2019; Benedetti et al. 2021; Wang et al. 2019a, 2020, 2021,

2022b, 2024a; Li et al. 2016, 2018, 2022, 2023). Two explanations may account for this phenomenon. On one hand, the intrinsic mechanism is the endosymbiosis (Kutschera and Niklas 2005). After a long-term genetic DNA exchange and evolution process driven by closely prey-predation interaction (Chen et al. 2012), more diversified phytoplankton probable responsible for subsequent higher tintinnid biodiversity in tropical compared to bipolar zones through endosymbiosis mechanism (Margulis and Sagan 2002; Clark et al. 2023).

On the other hand, physical barriers represent an extrinsic mechanism (Amargant-Arumí et al. 2024; Antoni et al. 2024; Chust et al. 2024). Biogeographically, the ecological connectivity of plankton biodiversity is hindered by large gyres or water masses within interconnected oceans (Yang et al. 2020), which are characterized by unique environmental sensitivities (Longhurst 2007). For instance, tintinnid species diversity exhibits distinct variations in the North Pacific Gyre, the Subarctic Gyre, and the Beaufort Gyre (Wang et al. 2020). Ultimately, understanding biodiversity dynamics across different

temperature zones will benefit for studying the future adaptation mechanisms of microzooplankton in response to climate change.

### 4.3 Physicochemical factors determine the habitat of microzooplankton

Hydrography habitat conditions formed by large gyres (horizontal) or water masses (vertical) are critical factors in reshuffling sophisticated species composition of microbial food web (Lennartz et al. 2024). Conventionally, temperature can

promote plankton biodiversity through regulating intrinsic temperature-dependent metabolic processes (Archibald et al. 2022; Lukić et al. 2022; Weisse 2024). Coincidentally, the statistically positive correlation observed between tintinnid species



richness and temperature (Figure 7) fully supports the abovementioned ecological process. In this perspective, we conclude that temperature determines organism mortality by affecting their thermal affinity within biogeochemical cycles (Stuart-Smith et al. 2015; Chust et al. 2024) through an indirect effect (Weisse and Sonntag 2016; Weisse 2024). Similarly, through

modulating osmotic pressure, salinity plays a crucial role in shaping the species composition of the microbial food web (Pedrós-Alió et al. 2000; Zang et al. 2024), and in hindering the dispersal of Pacific species into the Arctic Ocean (Wang et al. 2019b, 2022c). Our study, along with others, indicates that ciliate inhabiting higher salinity environments in both the TZ and NTZ (Figure 5) compared to bipolar regions might be a reflection of their higher osmotic pressure affinity.

Furthermore, due to their ecological role in the fundamental prey-predator interplay, Chl *a* (food items for upper trophic

levels) directly sustains the stability dynamics of marine ecosystems through both its quantity (abundance) and quality (content of unsaturated fatty acids) (Šolić et al. 2010; Våge and Thingstad 2015; Holm et al. 2022). Consequently, Chl *a* modulated the base of the entire marine ecosystem (Li et al. 2024). As direct micro-grazers of phytoplankton, both the abundance and species richness of ciliates exhibit a significant positive correlation with Chl *a* (Figure 5-7 and Figure S8), aligning with the aforementioned viewpoint regarding the ecological role of Chl *a*. As outlined above, considering the

significant correlation between hydrographic factors and ciliates (abundance and species richness) (Figure 7), coupled with the trait structure of ciliate susceptible to environmental change (Yu et al. 2022), we conclude that bottom-up control (prey availability, resource limitation) (Lu and Weisse 2022; Wang et al. 2023c, 2024c) plays a primary role compared to top-down control (limited by microcrustacean predators or top grazer) (Power 1992; Calbet et al. 2001; Worm and Myers 2003) for microzooplanktonic ciliate in the global marine ecosystem.

**4.4 Prediction for microzooplanktonic ciliate community to future global warming**

Global warming, primarily stemming from anthropogenic industrial-induced $CO_2$ emissions, have caused enduring and irreversible impacts on marine ecosystems globally, impelling a suite of threats to biodiversity and marine ecosystem, such as phenology evolution and adaptation, species poleward dispersal and body-size miniaturization (Daufresne et al. 2009; Poloczanska et al. 2013; Atkinson et al. 2015; Hastings et al. 2020; Møller and Nielsen 2020; Yasumiishi et al. 2020; Wang

and Wu 2022; Qian et al. 2023; Wang et al. 2024b). To date, existing facts indicate a significant increase in plankton abundance and species richness in bipolar and adjacent seas due to continuously rising temperature (Ershova et al. 2015; Wassmann et al. 2015; Hunt et al. 2016; Kim et al. 2020; Lewis et al. 2020; Mueter et al. 2021; Wang et al. 2022a, 2023a) within a short timeframe. However, it should be mentioned that future global warming is expected to induce species extirpations by both compelling species beyond their thermal limits (Benedetti et al. 2021) and disrupting optimal survival

habitats (Wang et al. 2024b).

Species poleward dispersal is another prominent aspect of plankton's responses to climate change (Hastings et al. 2020). Unfortunately, surface-dwelling ciliates (Kršinić 1982; Wang et al. 2019a, 2023a, 2024b) are particularly vulnerable to recent more frequent extreme temperature events, especially in tropical seas. Similarly, Benedetti et al. (2021) projected a median speed of approximately 35 km/decade for the poleward shift of species dispersal under a high $CO_2$ emission scenario

by the end of this century. In this perspective, our study provides a fundamental benchmark for understanding the adaptive strategies (extirpation, dispersal, or adaptation) of ciliate to rapid warming processes in global seas. Meanwhile, unlike "winner" pioneer species possessing strong adaptation abilities (Casoli et al. 2020; Boutin et al. 2023), native species characterized by lower adaptive ability, such as the Arctic endemic tintinnid species Ptychocylis urnula, may either migrate passively to new environments (Wang et al. 2022a, 2023a, 2024b) or collapsed by a combination of warming and

competition (Chust et al. 2024). Furthermore, the dynamics of future trophic food webs and biogeochemical flux in the global marine ecosystem will heavily rely on how indigenous and/or intrusive species adjust to a warmer ocean state amidst multiple ecosystem stressors.

## 5 Conclusions

Our results provides a comprehensive disparities in microzooplanktonic ciliate trait structure focused on size spectrum,

biodiversity, and biotic-abiotic interplay based on 175 stations across five temperature zones from the North Pole to the Southern Ocean (Antarctic). Concerning ciliate size spectrum, slope of the normalized abundance value displayed an inverse relationship between ciliate abundance and body-size, resembling a pyramid norm, while the biomass-size spectrum showed relatively smoother slopes. Steeper ciliate abundance size spectra slopes in bipolar seas compared to tropical, temperate, and sub-Arctic seas suggest a prevalent trend of miniaturization. Additionally, tintinnid biodiversity was highest in tropical and

subtropical seas and lowest in bipolar seas, likely influenced by endosymbiosis (intrinsic mechanism) and physical barriers (extrinsic mechanism). Furthermore, the interplay between biotic and abiotic factors manifested that temperature exert a primary influence on ciliate community structure. Under current foreseeable rapid global warming process, we conjecture that bottom-up control (resource limitation) playing a more primary role through an indreict way in the global marine ecosystem.

**Acknowledgements**

Special thanks to captains and crews of *R.V.* "Xuelong 2", "Xiangyanghong 01", "Xiangyanghong 10", "Dongfanghong 3" and "Kexue" for their great help in sampling periods during the cruises.

**Financial support**

This research was supported by the National Natural Science Foundation of China (42206258), the Shandong Provincial

Natural Science Foundation (ZR2022QD022), the National Natural Science Foundation of China (42276156; 42176228), and the International Research Project-Dynamics and Function of Marine Microorganisms (IRP-DYF2M): insight from physics and remote sensing, CNRS-CAS.



**Competing interests**

The contact author has declared that none of the authors has any competing interests.

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
