# Peer review of "Decoding pelagic ciliate (Ciliophora) community divergences in size spectrum, biodiversity and driving factors globally spanning five temperature zones"

_EGUsphere, 2024_

## Author Comment (AC1)

**Dear Editor**,

We finished the revision of the manuscript according to the questions and advices of the four reviewers. The following are the details of our responses (in blue color) to questions and advices of every reviewer.
The work of reviewers help improve the quality of the manuscript. We thank the thoughtful advice of the reviewers and hope the revision successfully answered the questions.
Best wishes

**Wuchang Zhang**

=========================================================
Reviewers' comments:
**Reviewer #1 (RC1)**:
The paper "Decoding Pelagic Ciliate (Protozoa, Ciliophora) Community Divergences in Size Spectrum, Biodiversity, and Driving Factors Spanning Five Global Temperature Zones" (Egusphere-2024-3888) studies pelagic ciliates across five temperature zones. Ciliates play a crucial role in the planktonic food web, and expanding our knowledge—especially through studies like this—is essential for understanding their future in the context of climate change.

The ciliate counting work is impressive. The paper is well-written, and the data analysis is highly appropriate. The discussion is engaging; however, some results should be explored in greater depth. Additionally, the discussion contains overly general ideas from the bibliography.

The main revisions should focus on the figures. In the paper, the figures are too small, and some are difficult to interpret. Some figures in the supplementary materials are more effective than those included in the main text. Please select the most appropriate figures to illustrate the results clearly.

For these reasons, I recommend this paper for publication with minor revisions.

Some mistakes:
1) Line 80: field
Response: We revised this previuos wrong word into "field" accordingly **in lines 79–80 in revised manuscript.**
Lines 79–80: By optimizing field observational data and available methods, this study aims to:…

2) Line 73, 87: ciliates
Response: We revised "ciliate" into "ciliates" accordingly **in lines 72–75 and lines 86–87 in revised manuscript.**

Lines 72–75: As grazer of pelagic phytoplankton, response of microzooplanktonic ciliates to ocean warming in the bipolar and adjacent seas is substantial (Li et al. 2022; Wang et al. 2022a, 2023a, 2023b, 2024b), yet comparative assessments amid their trait structure (e.g., size spectra, biodiversity and biotic-abiotic interplay) remain unexplored to date.

Lines 86–87: Based on their latitudinal locations, field samplings of microzooplanktonic ciliates were conducted in five temperature zones (Trewartha et al. 1967).

3) Line 108: no space after (Utermöhl 1958)

Response: We revised accordingly **in lines 105–106 in revised manuscript.**

Lines 105–106: After two rounds of siphon process, a final of 25 mL highly concentrated sample was obtained, and then settled in a Utermöhl counting chamber (Utermöhl 1958).

4) Line 141: we used

Response: We revised accordingly **in lines 141–143 in revised manuscript.**

Lines 141–143: In the following, based on the slope condition, we used the decreasing rate ($\Delta_D$) or increasing rate ($\Delta_I$) according to ciliate abundance or species richness and environmental variables to quantize their interplay in the global seas.

5) Figure 5 and figure 6: legend for a, b, and c

Response: We added legend for previous Figure 5 and Figure 6, and moved these two figures into present Figure S8 and Figure S9 **in revised supplementary material in revised manuscript.**

**Figure S8** (previous Figure 5): Variations in slopes between ciliate abundance and temperature (a)/salinity (b)/Chl *a* (c) at discrete depth in each temperature zone.

**Figure S9** (previous Figure 6): Variations in slopes between tintinnid species richness and temperature (a)/salinity (b)/Chl *a* (c) at discrete depth in each temperature zone.

6) Line 368: indirect

Response: We revised this previuos wrong word into "indirect" accordingly **in lines 360–361 in revised manuscript.**

Lines 360–361: Under current foreseeable rapid global warming process, we conjecture that bottom-up control (resource limitation) playing a more primary role through an indirect way in the global marine ecosystem.

7) The ciliate counting work is impressive. The paper is well-written, and the data analysis is highly appropriate. The discussion is engaging; however, some results should be explored in greater depth. Additionally, the discussion contains overly general ideas from the bibliography.

Response: Thank you for your appreciation. We realized that some results indeed should be explored in greater depth, and the discussion contains overly general ideas from the bibliography, thus we revised the whole discussion part to fit the scope of

this manuscript accordingly **in revised manuscript**.

8) The main revisions should focus on the figures. In the paper, the figures are too small, and some are difficult to interpret. Some figures in the supplementary materials are more effective than those included in the main text. Please select the most appropriate figures to illustrate the results clearly.

Response: We realized that the figures are too small in the manuscript. After careful consideration, we revised previous "Figure 3" into present "Figure 3 and Figure 4". In addition, we moved previous "Figures 5–6" into present "Figures S8–S9" (Supplementary material) accordingly **in revised manuscript.**

---

## Author Comment (AC2)

**Dear Editor**,

We finished the revision of the manuscript according to the questions and advices of the four reviewers. The following are the details of our responses (in blue color) to questions and advices of every reviewer.

The work of reviewers help improve the quality of the manuscript. We thank the thoughtful advice of the reviewers and hope the revision successfully answered the questions.

Best wishes

**Wuchang Zhang**

============================================================

**Reviewer #2 (CC1)**: The data in this paper are obtained through the accumulation of several scientific cruises, which is very precious and rare. The differences in abundance, biomass, diversity and size spectrum of pelagic ciliate among the five temperature zones were demonstrated through measured data, in addition to the differences between latitudes, the data also showed differences in the vertical patterns of planktonic ciliate abundance, biomass and size structure in the five temperature zones, which is of great value for understanding the global distribution of pelagic ciliate.

1) The analysis on biotic-abiotic interplay is also very meaningful, but the current analysis results in 3.4 are somewhat confused. The main reason is that it is not necessary to show the abiotic factors controlling the spatial variation of ciliates within each temperature zone, because these results are determined by the range of sampling stations in each temperature zone, and are independent of the comparison between the five temperature zones. Thus, the content of part 3.4 and the corresponding discussion needs to be adjusted. I suggest that PCA focus on analyzing the relationship between the dominant species in the five temperature zones and various abiotic factors.

Response: In order to delete confusion, we moved previous "Figures 5–6" into present "Figures S8–S9" (Supplementary material) accordingly in revised manuscript. Meanwhile, we deleted several sentences to better exhibiting the biotic–abiotic interplay. Regarding the PCA analysis, we want to find out the role of environmental factors played in ciliate composition (both ciliate abundance and species richness) in each temperature zone. The relationship between the dominant species in the five temperature zones and various abiotic factors might be have minimal correlation due to the range of sampling stations in each temperature zone. Consequently, the strategists we have adopted were compared the internal correlation among each temperature zone.

2) Lines 149-150: "At 200 m depth, temperature and Chl a peaked in the TZ and North Frigid Zone (NFZ), respectively, deviating from salinity patterns, which exhibited high values in both the TZ and NFZ" The expression of this sentence is not

clear, modify it to make it clearer.

Response: We revised this sentence accordingly **in lines 149–150 in revised manuscript.**

Lines 149–150: At 200 m depth, temperature peaked in the TZ and Chl a peaked in the North Frigid Zone (NFZ), contrasting with salinity patterns, which displayed high values in both the TZ and NFZ (Figure 2 and Figure S1).

3) Lines 142-144: The vertical distribution of chlorophyll a in SAZ is not described.

Response: We added the vertical distribution of Chl *a* in SAZ **in lines 150–152 in revised manuscript.**

Lines 150–152: Vertically, both temperature and Chl a declined in the NFZ and Sub-Arctic Zone (SAZ) (surface-peak pattern), while salinity increased from the surface to 200 m layers across all regions (Figures S1–S3).

4) Lines 170-171: "Vertically, the large (> 50µm) and small size-fractions exhibited an inverse distribution characteristic across five temperature zones"The meaning of this sentence is not clear, modify it to make it clearer.

Response: We revised this sentence accordingly **in lines 170–172 in revised manuscript.**

Lines 170–172: Vertically, the relative abundance of the large size-fraction (>50 µm) exhibited a decreasing trend, whereas the small size-fraction displayed an increasing trend across the five temperature zones (Figures S5).

5) Lines 209-240: The large differences in the relationship between biological and abiotic organisms in different temperature zones may be mainly caused by the difference in the selection of sampling areas, rather than the fundamental differences between temperature zones.

Response: We hold the similar viewpoint that the large differences in the relationship between biological and abiotic organisms in different temperature zones may be mainly caused by the difference in the selection of sampling areas, rather than the fundamental differences between temperature zones. Therefore, the strategists we have adopted were compared the internal correlation among each temperature zone at specific sampling depth (0, 50, 100, and 200 m). In Figure S10, we just want to find out the linear relation between ciliate and each environmental fctor at all sampling depth among each temperature zone.

6) Lines 278-279: "the general trend of steeper slopes at the surface compared to the 200 m layer across all regions suggests a community size shift influencing carbon flux efficiency towards higher trophic levels"It is difficult to understand the relationship between the half sentence before and the half sentence after "suggest", and additional explanation is needed.

Response: In order to make this sentence more clear, we added an additional explanation accordingly **in lines 269–272 in revised manuscript.**

Lines 269–272: Furthermore, the consistently steeper slopes at the surface compared

to the 200 m layer across all regions (Figure 3) suggest: (1) a depth-dependent shift in pelagic ciliate community size structure, and (2) greater accessibility of prey for meso-/macro-zooplankton in surface waters compared to the 200 m layer, thereby influencing carbon flux efficiency to higher trophic levels (Stukel et al., 2024).

7) Lines 291-292: "the steeper slopes observed in the abundance size spectra in the bipolar seas compared to the tropical, temperate, and sub-Arctic seas might reflect a prevailing trend towards miniaturization"also, it is difficult to understand the relationship between the half sentence before and the half sentence after "might reflect", and additional explanation is needed.

Response: At present, we find out the phenomenon that the steeper slopes observed in the abundance size spectra in the bipolar seas compared to the tropical, temperate, and sub-Arctic seas, but to be honest, it's hard for us to explore the explanation. Thus we deleted this sentence **in revised manuscript.**

---

## Author Comment (AC3)

**Dear Editor**,

We finished the revision of the manuscript according to the questions and advices of the four reviewers. The following are the details of our responses (in blue color) to questions and advices of every reviewer.

The work of reviewers help improve the quality of the manuscript. We thank the thoughtful advice of the reviewers and hope the revision successfully answered the questions.

Best wishes

**Wuchang Zhang**
============================================================

**Reviewer #3 (CC2)**: The authors presented a detailed and comprehensive dataset of ciliate community distribution across the major temperature zones in the sea, and the ciliate morphospecies were identified in 1,117 samples taken at 175 stations in the Arctic and sub-Arctic Ocean, the North Pacific, the tropical western Pacific, the Indian Ocean, and the Southern Ocean (in global scale). Meanwhile, ciliate abundance and biomass size spectra, as well as species richness and diversity, were related to environmental parameters and depth. Objectives and rationales are clear, robust and well presented. Furthermore, the authors' analyses confirm general trends (e.g., size-diversity and temperature-diversity relationships for aloricate ciliates and tintinnids, a decrease of ciliate abundance and biomass with depth) and present numerous details for each biogeographic zone worth publishing. However, several shortcomings should be reviewed to more fill the scope of their overall goal. In conclusion, I recommend this manuscript for publication in the Ocean Science characterized with high-ranked international journal after revising some specific comments as follow.

Specific comments:
1) Title: pelagic ciliates belonged to Protozoa is well-known in marine plankton realm, thus it's no need to strengthen it in the title. Just delete this term.
Response: We deleted "Protozoa" accordingly **in revised manuscript.**

2) line 48: Common sense error. The "anthropogenic CO2 emissions" should be revised into "anthropogenic CO2 emissions".
Response: We revised into "anthropogenic $CO_2$ emissions" accordingly **in lines 47–48 in revised manuscript.**
Lines 47–48: Over recent decades, anthropogenic $CO_2$ emissions have led to increased atmospheric concentrations and greater global radiative forcing (Tagliabue et al. 2023),…

3) line 93: please make sure that whether the cruise conducted in the Indian Ocean in March 2021 aboard the R.V. "Xiangyanghong 10"? I remembered that this cruise

might be conducted by the R.V. "Xiangyanghong 6" in previous manuscript I have reviewed.

Response: After carefully checking, we revised into *R.V.* "Xiangyanghong 6" **in lines 90–92 in revised manuscript.**

Lines 90–92: 4, the Torrid Zone (TZ), which includes the tropical western Pacific in December 2016 and August 2017 aboard the *R.V.* "Kexue", and the Indian Ocean in March 2021 aboard the *R.V.* "Xiangyanghong 6".

4) The Methods section lacks detail. I recognized that the method how you calculated the size-fraction of aloricate ciliate, while how the biomass spectra were constructed (size categories?) is unclear. Please state clearly relate to the calculation of the biomass spectra.

Response: We added the calculation of size spectra biomass and revised this sentence accordingly **in lines 122–123 in revised manuscript.**

Lines 122–123: Concerning size spectra biomass, ciliate biomass were calculated based their specific organism volume and conversion equation, then    categorized into each size spectrum as in Wang et al. (2024b).

5) line 122: Convert pg C to µg C.

Response: We accepted suggestions and revised into "$0.19 \times 10^{-6}$ µg C µm$^{-3}$" **in lines 120–121 in revised manuscript.**

Lines 120–121: Additionally, a conversion factor ($0.19 \times 10^{-6}$ µg C µm$^{-3}$) was used for calculating aloricate ciliate carbon biomass (Putt and Stoecker 1989).

6) line 153: What do you mean the "sandwich structure" for temperature. I cannot find this phenomenon clearly in Figures S1 and S3. Therefore replace it.

Response: We accepted suggestions and revised into "low–high–low structure" **in lines 152–154 in revised manuscript.**

Lines 152–154: Moreover, temperature displayed a low–high–low structure at inner stations of the South Frigid Zone (SFZ), and Chl *a* peaked at subsurface layers in both the North Temperate Zone (NTZ) and TZ (Figures S1 and S3).

7) Figure 2: you have mentioned the abbreviation of the five temperature zones in figure 1: the North Frigid Zone (NFZ), sub-Arctic Zone (SAZ), North Temperate Zone (NTZ), Torrid Zone (TZ) and South Frigid Zone (SFZ), thus there is no need to write this part again.

Response: We accepted suggestions and revised accordingly **in line 174 in revised manuscript.**

Line 174: Figure 2: Variations in environmental variables and ciliate abundance and biomass at discrete depth in each temperature zone.

8) line 275: I wondered that it not clear what is meant by "monospecific trophic levels, such as microzooplanktonic ciliates"; ciliates represent more than one trophic level (i.e., as phototrophs, bacterivores, herbivores/omnivores, predators, parasites). Please

state it clearly in this part.

Response: At this part, we just focused on one group of microzooplanktonic ciliates, thus the words of "monospecific trophic levels" was unseemliness. Based on our viewpoint, we revised into "specific zooplankton assemblage" **in lines 266–267 in revised manuscript.**

Lines 266–267: Currently, research on specific zooplankton assemblage, such as microzooplanktonic ciliates (Wang et al. 2024a), is rarely studied on a global scale.

9) In the discussion part, the author mentioned that the bottom-up control is the resource limitation as previous pointed. In this study, temperature is environmental factor (=environmental filter) for which exert a primary influence..... I strongly suggest to clearly separate in Discussion the interpretation of environmental filters and trophic mechanisms as explanatory variables for the patterns revealed and to make corresponding corrections in the Abstract.

Response: We accepted suggestions and separated the interpretation of environmental filters and trophic mechanisms in the Abstract **in lines 29–31 and lines 317–328 in revised manuscript.**

Lines 29–31: Moreover, a multivariate biota-environment analysis indicated that temperature exert a primary influence on ciliate community constitution in the global marine ecosystem, and the bottom-up control play a key role in shaping assemblages.

Lines 317–328: Furthermore, the Chl $a$ functionally serves as a critical ecological mediator in marine food webs, influencing ecosystem stability through both quantitative (abundance) and qualitative (polyunsaturated fatty acid composition) pathways via the fundamental prey-predator interplay (Šolić et al. 2010; Våge and Thingstad 2015; Holm et al. 2022). Consequently, Chl $a$ modulated the energy flow of the entire marine ecosystem (Li et al. 2024). As direct micro-grazers of phytoplankton, both the abundance and species richness of ciliates exhibit a significant positive correlation with Chl $a$ (Figure 6 and Figures S8–S10), aligning with the aforementioned viewpoint regarding the ecological role of Chl a. As outlined above, coupled with our results about multivariate analyses revealed strong hydrographic-ciliate relationships (Figure 6), while observed trait plasticity in ciliate communities (Yu et al. 2022) further supports the predominance of bottom-up control mechanisms (resource availability, prey quality) (Lu and Weisse 2022; Wang et al. 2023c, 2024c) over top-down regulation (predation pressure from microcrustaceans) (Power 1992; Calbet et al., 2001; Worm and Myers, 2003) in structuring global microzooplankton communities. This trophic cascade pattern underscores the fundamental role of primary production dynamics in governing ciliate population ecology across marine ecosystems.

10) In section 4.3, a recent meta-analysis contradicts the authors' conclusion because ciliate mortality appears to be unaffected by temperature (Weisse, 2024, Limnol. Oceanogr.), which was inconsistent with your results. How do you cope with this phenomenon? By the way, T determines organism mortality contradicts empirical evidence for ciliates (Weisse 2024)

Response: We studied carefully about the recent meta-analysis that ciliate mortality appears to be unaffected by temperature (Weisse, 2024). Regarding this phenomenon, majority previous studies manifested that temperature emerges as a principal driving factor of plankton composition and dispersal, particularly in high-latitude polar regions, due to its direct impact on physiological processes (e.g., respiration, productivity, reproduction) via thermally dependent metabolic regulation (e.g., Knies et al., 2009; Stuart-Smith et al. 2015; Archibald et al., 2022; Chust et al. 2024). In addition, temperature determine the habitat conditions for pelagic plankton. Therefore, we approved the viewpoint that ciliate mortality affected by temperature. We also revised accordingly **in lines 317–328 in revised manuscript.**

**References**

Knies, J. Kingsolver, J. and Burch, C.: Hotter is better and broader: Thermal sensitivity of fitness in a population of bacteriophages. Am. Nat. 173, 419–430, doi:10.1086/597224, 2009.

Stuart-Smith, R. Edgar, G. Barrett, N. Kininmonth, S. and Bates, A.: Thermal biases and vulnerability to warming in the world's marine fauna, Nature 528, 88–92, doi:10.1038/nature16144, 2015.

Archibald, K. Dutkiewicz, S. Laufkötter, C. and Moeller, H.: Thermal responses in global marine planktonic food webs are mediated by temperature effects on metabolism, J. Geophys. Res. Oceans 127, e2022JC018932, doi:10.1029/2022JC018932, 2022

Chust, G. Villarino, E. McLean, M. Mieszkowska, N. Benedetti-Cecchi, L. Bulleri, F. Ravaglioli, C. Borja, A. Muxika, I. Fernandes-Salvador, J.… and Lindegren, M.: Cross-basin and cross-taxa patterns of marine community tropicalization and deborealization in warming European seas, Nat. Commun. 15, 2126, doi:10.1038/s41467-024-46526-y, 2024.

11) line 316: the author mentioned that the positive correlation between tintinnid species richness and temperature, while this correlation may be an indirect effect.

Response: Our study revealed that tintinnid species richness and temperature had significant positive correlation through biotic-abiotic analysis. This phenomenon was fit the viewpoint that temperature can promote plankton biodiversity through regulating intrinsic temperature-dependent metabolic processes. Regarding this problem, we agreed with the reviewer's point that correlation may be an indirect effect between pelagic ciliate and temperature (Weisse and Sonntag 2016; Weisse 2024). We also revised this sentence accordingly **in lines 309–312 in revised manuscript.**

Lines 309–312: In this perspective, we conclude that temperature determines organism mortality by affecting their thermal affinity within biogeochemical cycles (Knies et al., 2009; Stuart-Smith et al. 2015; Archibald et al., 2022; Chust et al. 2024) through an indirect effect (Weisse and Sonntag 2016; Weisse 2024).

12) line 349: $CO_2$

Response: We revised into "$CO_2$" **in lines 342–344 in revised manuscript.**

Lines 342–344: Similarly, Benedetti et al. (2021) projected a median speed of approximately 35 km/decade for the poleward shift of species dispersal under a high $CO_2$ emission scenario by the end of this century.

13) line 360: add the total number of samples.

Response: We added the total number of samples accordingly **in lines 353–355 in revised manuscript.**

Lines 353–355: Our results provides a comprehensive disparities in microzooplanktonic ciliate trait structure focused on size spectrum, biodiversity, and biotic-abiotic interplay based on 1117 water samples from 175 stations across five temperature zones from the North Pole to the Southern Ocean (Antarctic).

14) At last, I'm curious about a phenomenon that the author spend a lot of description in discussing the relationship between the environmental variables and "bottom-up control", and previous studies recognized that the plankton community was strict restricted by outer environmental resources, which was known as "bottom-up control". However, how do you identify the correlation between the environmental variables and "bottom-up control"?

Response: The bottom-up control refers to an ecological mechanism where lower trophic levels (e.g., nutrients, primary producers) regulate the structure and productivity of higher trophic levels (e.g., zooplankton, fish) in marine ecosystems (Lu and Weisse 2022). In other words, bottom-up control can be regarded as a resource-limited environment. In the marine ecosystem, environmental variables play a key role in reshuffling sophisticated species composition of microbial food web (Lennartz et al. 2024), such as temperature determines organism mortality through modulating their thermal affinity within biogeochemical cycles; Chl *a* directly sustains the stability dynamics of upper trophic levels through providing food items in predation process. Therefore, we consider that the environmental variables and "bottom-up control" are inseparable factors during biotic-abiotic interplay.

**References**:

Lu, X. and Weisse, T.: Top-down control of planktonic ciliates by microcrustacean predators is stronger in lakes than in the ocean, Sci. Rep. 12, 10501, doi:10.1038/s41598-022-14301-y, 2022.

Lennartz, S. Keller, D. Oschlies, A. Blasius, B. and Dittmar, T.: Mechanisms underpinning the net removal rates of dissolved organic carbon in the global ocean, Global Biogeochem. Cy. 38, e2023GB007912, doi:10.1029/2023GB007912, 2024.

---

## Author Comment (AC4)

**Dear Editor**,

We finished the revision of the manuscript according to the questions and advices of the four reviewers. The following are the details of our responses (in blue color) to questions and advices of every reviewer.

The work of reviewers help improve the quality of the manuscript. We thank the thoughtful advice of the reviewers and hope the revision successfully answered the questions.

Best wishes

**Wuchang Zhang**
==============================================================

**Reviewer #4 (CC3)**: The manuscript "Decoding pelagic ciliate (Protozoa, Ciliophora) community divergences in size spectrum, biodiversity abd driving factors spanning global five temperature zones" by Wang and collaborators uses an impressive dataset on the distribution of pelagic ciliates over different ecological regions, describing important community features such as size and species composition using environmental parameters to contextualize their findings. While the data is fantastic and should be published, the current version of the manuscript still needs further work. I provide some specific comments below:

1) The language should be revised and the text can be streamlined in several parts. A clear example is the title, which is rather long and not really informative.
Response: We accepted suggestions and revised the whole manuscript accordingly. As for title, it conveyed the three main themes of this manuscript, thus we revised into "Decoding pelagic ciliate (Ciliophora) community divergences in size spectrum, biodiversity and driving factors spanning global five temperature zones" **in revised manuscript.**

2) Considering that you target only one planktonic group, maybe the normalized size spectra approach is not the best to describe the variation in sizes (as also discussed by the authors in the manuscript). Could simple metrics, such as the average size be more informative?
Response: We agreed with your viewpoint that the normalized size spectra approach is not the best to describe the variation in sizes for only one planktonic group. While regarding different temperature zones, there were several variations for pelagic ciliates lived in oceanic habitat. We tried to find out their divergences in size spectra aspect. Actually, we used the average size of each ciliate size-fraction (for instance, we used the 15 μm size-fraction in 10–20 μm size-fraction) in size spectrum analysis in the manuscript. We also revised accordingly **in lines 114–115 in revised manuscript.**
Lines 114–115: Furthermore, we select the average value (15, 25, 35, 45 μm,…, etc) of each size-fraction of both loricate ciliate and tintinnid as the counting criterion for

ciliate size spectra (Wang et al. 2024b).

3) The authors should also consider other traits than size to describe the communities, such as the presence/absence of lorica and trophy mode could be more meaningfull than the normalized size spectra.

Response: Dear reviewer, thank you very much for proposing these valuable suggestions (the presence/absence of lorica and trophy mode) for pelagic ciliate trait study. To be honest, we counted the presence/absence of tintinnid lorica only in recent two years (starting from 2023 in the Arctic Ocean). Thus relative data was not recorded in the Bering Sea, North Pacific, tropic western Pacific and Indian Ocean. Therefore, we can not conduct this trait structure. Concerning trophy mode, pelagic ciliate belonged to the top grazer of the microbial food web. To date, we already start to study its role in the microbial food web, and relative study in tropic western Pacific have been published in Marine Pollution Bulletin. The other one relate to the Arctic Ocean just submitted in the Global Biogeoscience Cycles. In the near future, we will put more emphasis on uncovering trophy mode of pelagic ciliate in marine ecosystem.

4) I have reservations about how the statistical methods were used by the authors. A constrained ordination using the entire data set might be more appropriate than the ordination analysis. In addition, the relationship between the community and enviromental variables could be done with a more comprehensive model (e.g. GLM that also includes zode and depth as independent variables).

Response: Dear reviewer, we appreciate for your valuable advice that using the entire data set might be more appropriate than the ordination analysis. To be honest, we hold the similar viewpoint with reviewer 2 that the large differences in the relationship between biological and abiotic organisms in different temperature zones may be mainly caused by the difference in the selection of sampling areas, rather than the fundamental differences between temperature zones. Therefore, the strategists we have adopted were compared the internal correlation among each temperature zone at specific sampling depth (0, 50, 100, and 200 m). In Figure S10, we just want to find out the linear relation between ciliate and each environmental fctor at all sampling depth among each temperature zone. Additionally, we are really sorry that we did not conduct a comprehensive GLM model due to our complex data in both latitudinal and vertical directions, thus we have no idea on how we conduct this model.

5) Considering that seasonality is also important to modulate protozoan communities, are all the datasets comparable in this regard?

Response: We acknowledged that seasonality is important to modulate protozoan communities, but this phenomenon was obvious in both temperate and polar seas. Regarding tropic seas in both the Pacific and Indian Ocean, the community structure including vertical distribution pattern, abundance and biomass values, species composition were almost same (e.g., Sohrin et al., 2010; Li et al., 2018; Wang et al., 2019a, 2020, 2022b). In other words, seasonality might not be a driving factor for pelagic ciliate community in tropic seas.

**References**:

Sohrin, R. Imazawa, M. Fukuda, H. and Suzuki, Y. Full–depth profiles of prokaryotes, heterotrophic nanoflagellates, and ciliates along a transect from the equatorial to the subarctic central Pacific Ocean. Deep–Sea Res. II 57, 1537–1550. doi:10.1016/j.dsr2.2010.02.020, 2010.

Li, H. Zhang, W. Zhao, Y. Zhao, L. Dong, Y. Wang, C. Liang, C. and Xiao, T.: Tintinnid diversity in the tropical West Pacific Ocean, Acta Oceanol. Sin. 37, 218–228, doi:10.1007/s13131-018-1148-x, 2018.

Wang, C. Li, H. Zhao, L. Zhao, Y. Dong, Y. Zhang, W. and Xiao, T.: Vertical distribution of planktonic ciliates in the oceanic and slope areas of the western Pacific Ocean, Deep-Sea Res. II 167, 70–78, doi:10.1016/j.dsr2.2018.08.002, 2019a.

Wang, C. Li, H. Xu, Z. Zheng, S. Hao, Q. Dong, Y. Zhao, L. Zhang, W. Zhao, Y. and Xiao, T.: Difference of planktonic ciliate communities of the tropical West Pacific, the Bering Sea and the Arctic Ocean, Acta Oceanol. Sin. 39, 9–17, doi:10.1007/s13131-020-1541-0, 2020.

Wang, C. Li, H. Dong, Y. Zhao, L. Grégori, G. Zhao, Y. Zhang, W. and Xiao, T. Planktonic ciliate trait structure variation over Yap, Mariana and Caroline seamounts in the tropical western Pacific Ocean. J. Oceanol. Limnol. 39, 1705–1717. doi:10.1007/s00343-021-0476-4, 2021.

Wang, C. Zhao, Y. Du, P. Ma, X. Li, S. Li, H. Zhang, W. and Xiao, T.: Planktonic ciliate community structure and its distribution in the oxygen minimum zones in the Bay of Bengal (eastern Indian Ocean), J. Sea Res. 190, 102311, doi:10.1016/j.seares.2022.102311, 2022b.

6) Some of the sampling campaigns occurred over relatively large areas, which could be sampling over systems divided by ocranographyc features, such as fronts. Was the intra-zone variability taken into account?

Response: We awared that sampling campaigns occurred over relatively large areas and the marine pelagic ciliate community exhibited some differences by oceanographic features. Actually, the intra-zone variability was taken into account during writing process. In a whole, compared to different temperature zones, we found that the intra-zone variability of ciliate community was negligible.

---

## Author Response (AR2)

**Dear Editor**,

We finished the revision of the manuscript according to the questions and advices of the reviewer. The following are the details of our responses (in blue color) to questions and advices of the reviewer.

The work of reviewers help improve the quality of the manuscript. We thank the thoughtful advice of the reviewer and hope the revision successfully answered the questions.

Best wishes

**Wuchang Zhang**
=============================================================
Public justification (visible to the public if the article is accepted and published):
Dear Dr. Wang and co-authors,

Your revised manuscript has much improved. Both referees are satisfied with your response to their reviews.

I have read your revised manuscript and come to a list of comments below. First, there are three comments by a referee (first three in my listing below) that I think should also find a way into the manuscript.

My impression of the manuscript is that the data and results are great. I think there is more discussion in all of these data. You could go deeper into these and I hope and encourage you to expand the discussion to some extent.

With kind regards
Mario Hoppema

=============================================================
**Reviewers' comments**:
1.  The third point by referee #4 (CC3):
"3) The authors should also consider other traits than size to describe the communities, such as the presence/absence of lorica and trophy mode could be more meaningfull than the normalized size spectra."
I agree with the referee. Your answer to the comment is agreed, but you should also mention these points in the manuscript.
Response: We added the information about the presence/absence of lorica and trophy mode accordingly **in lines 113–114 and lines 267–268 in revised manuscript.**
Lines 113–114: Moreover, we did not distinguish the presence/absence of tintinnid lorica during the sample counting process.
Lines 267–268: Additionally, more emphasis should be put on uncovering trophy mode of pelagic ciliate in marine ecosystem.

2. The fourth comment by referee #4 (CC3):

"4) I have reservations about how the statistical methods were used by the authors. A constrained ordination using the entire data set might be more appropriate than the ordination analysis."

Again, your answer is agreed, but I would like to find it in the manuscript as well.

Response: We added the information accordingly **in lines 141–144 in revised manuscript.**

Lines 141–144: Moreover, in order to reduce deviation in the relationship between biological and abiotic in different temperature zones may be mainly caused by the difference in the selection of sampling areas, rather than the fundamental differences between temperature zones, the internal correlation among each temperature zone at specific sampling depth (0, 50, 100, and 200 m) were compared in the following text.

3. The fifth comment by referee #4 (CC3):

"5) Considering that seasonality is also important to modulate protozoan communities, are all the datasets comparable in this regard?"

Again, please add this info to the manuscript.

Response: We added the relevant information about seasonality and references accordingly **in lines 135–138 in revised manuscript.**

Lines 135–138: Finally, although seasonality is important to modulate protozoan communities, but this phenomenon was obvious in both temperate and polar seas. Regarding tropic seas in both the Pacific and Indian Ocean, the community structure including vertical distribution pattern, abundance and biomass values, species composition were almost same (e.g., Sohrin et al., 2010; Li et al., 2018; Wang et al., 2019a, 2020, 2022b).

4. Title: I think a slight modification (globally vs. global) would make it better: Decoding pelagic ciliate (Ciliophora) community divergences in size spectrum, biodiversity and driving factors globally spanning five temperature zones

Response: We accepted suggestions and revised the title accordingly **in lines 1–3 in revised manuscript.**

Lines 1–3: Decoding pelagic ciliate (Ciliophora) community divergences in size spectrum, biodiversity and driving factors globally spanning five temperature zones.

5. L26-27 "Moreover, ciliate size spectra exhibited a decrease trend from small to large size spectra, with steeper slopes observed in bipolar zones (NFZ and SFZ) compared to the other temperature zones." This sentence is not clear. A decreasing trend from small to large sizes spectra is confusing and an arbitrary reader does not understand this. And which slope is meant here, which is getting steeper? This needs more explanation.

Response: In order to make this sentence more clearly, we revised "slope" (data of the size spectrum at specific temperature zone) into "slope line" (comparable tendency for evaluating the decreasing trend from small to large size spectrum) accordingly **in lines 26–28 in revised manuscript.**

Lines 26–28: Moreover, although abundance of ciliate size spectra exhibited a decrease trend from small to large size spectra globally, the steeper slope lines observed in both polar zones (NFZ and SFZ) compared to the other temperature zones.

6. L27 "in bipolar zones", change to: in both polar zones
Response: We revised the previous "in bipolar zones" into "in both polar zones" accordingly **in lines 26–28 in revised manuscript.**
Lines 26–28: Moreover, although abundance of ciliate size spectra exhibited a decrease trend from small to large size spectra globally, the steeper slope lines observed in both polar zones (NFZ and SFZ) compared to the other temperature zones.

7. L29 "while bipolar seas", change to: while the polar seas
Response: We revised the previous "while bipolar seas" into "while the polar seas" accordingly **in lines 28–29 in revised manuscript.**
Lines 28–29: Latitudinally, ciliate abundance and tintinnid biodiversity exhibited an anti-phase relationship, where the TZ hosted peak biodiversity while the polar seas showed the highest abundance.

8. L30 "exert a primary influence on ciliate community constitution" But how does it exert influence?
Response: We revised "exert" into "play" accordingly **in lines 29–31 in revised manuscript.**
Lines 29–31: Furthermore, a multivariate biota-environment analysis indicated that temperature play a primary influence on ciliate community constitution in the global marine ecosystem, and the bottom-up control play a key role in shaping assemblages.

9. L32-33 "can be generalised for assessing the potential effects of climate change on pelagic microzooplankton in future marine realm." Is there any evidence for this in the paper? If not, then this contention should be toned down.
Response: We revised previous "microzooplankton" into "ciliates" to toned down the contention accordingly **in lines 31–33 in revised manuscript.**
Lines 31–33: In conclusion, these results underscore the unprecedented divergences in ciliate trait structure among five temperature zones and can be generalised for assessing the potential effects of climate change on pelagic ciliates in future marine realm.

10. L38-41 A verb is missing in this sentence. Therefore, insert exists: Albeit a myriad of prevailing research exists relevant to …
Response: We accepted suggestions and revised accordingly **in lines 38–41 in revised manuscript.**
Lines 38–41: Albeit a myriad of prevailing research exists relevant to plankton biogeography and its interplay with environmental drivers highlighting its importance

in disentangling marine ecosystems and biogeochemical cycles (e.g., Wang et al. 2020; Darnis et al. 2022; Segaran et al. 2023; Tagliabue et al. 2023).

11.  L48 Tagliabue et al. 2023 is not the correct reference at this place.
Response: We changed "Tagliabue et al. 2023" into "Singh et al. 2021" accordingly **in lines 47–48 in revised manuscript.**
Lines 47–48: Over recent decades, anthropogenic $CO_2$ emissions have led to increased atmospheric concentrations and greater global radiative forcing (Singh et al. 2021), triggering diverse ecological feedbacks worldwide….

12. L52 polar instead of bipolar
Response: We revised "bipolar" into "polar" accordingly **in lines 51–53 in revised manuscript.**
Lines 51–53: In this sense, extensive existing studies put emphasis on biotic community response to climate change in the polar and adjacent seas owing to their higher susceptibility compared to tropical, subtropical, and temperate seas.

13.  L68 "holopelagic species that project the relevant adaptative strategies" It is not clear what project of strategies implied? Please explain and use different wording.
Response: We revised this sentence accordingly **in lines 67–68 in revised manuscript.**
Lines 67–68: Recent escalation in global warming have imposed a cascade of impacts on aquatic ecosystems, presenting a formidable challenge to inherent holopelagic species that modify their relevant adaptative strategies.

14. L73 polar instead of bipolar
Response: We revised "bipolar" into "polar" accordingly **in lines 72–73 in revised manuscript.**
Lines 72–73: As grazer of pelagic phytoplankton, response of microzooplanktonic ciliates to ocean warming in the polar and adjacent seas is substantial.

15. L78 delete sophisticated, as it does not fit here
Response: We accepted suggestions and deleted this word accordingly **in lines 76–78 in revised manuscript.**
Lines 76–78: Consequently, elucidating microzooplanktonic ciliate size spectra, species diversity and biotic-abiotic interplay at a global-scale is critical for projecting future marine ecosystem dynamics, particularly given their unresolved role in plankton response to climate changes.

16. L82 delete process, as it is not necessary here
Response: We deleted this word accordingly **in lines 81–83 in revised manuscript.**
Lines 81–83: Given the current foreseeable rapid climate change, this study will offer a benchmark for facilitating the phenological and bioclimatic progression of microzooplankton shifts in future global marine ecosystem realm.

17. L82 benchmark instead of valuable norm

Response: We accepted suggestions and revised into "benchmark" **in lines 81–83 in revised manuscript.**

Lines 81–83: Given the current foreseeable rapid climate change, this study will offer a benchmark for facilitating the phenological and bioclimatic progression of microzooplankton shifts in future global marine ecosystem realm.

18. L95 Please give the precision of the measurements of temperature, salinity and chlorophyll a in vivo fluorescence.

Response: Dear reviewer, to be honest, we did not know the precision of the measurements of temperature, salinity and chlorophyll *a* in vivo fluorescence by the CTD (SeaBird SBE 911). After each cruise, all above-mentioned environmental data were recorded by the CTD during its combined sampling process.

19. Figure 1 caption: polar instead of bipolar. Add the definition of AO used in the figure

Response: We accepted suggestions and revised "bipolar" into "polar" **in lines 101–102 in revised manuscript.** In addition, we revised "AO" into "Arctic Ocean" **in revised Figure 1 in revised manuscript.**

Lines 101–102: Figure 1: Survey stations and transects (Tr.) in the tropical, temperate and polar seas. NFZ, North Frigid Zone; SAZ, Sub-Arctic Zone; NTZ, North Temperate Zone; TZ, Torrid Zone; SFZ, South Frigid Zone.

20. L120 Please use present tense here: C is the …, V is the…

Response: We accepted suggestions and revised accordingly **in line 121 in revised manuscript.**

Line 121: Where $C$ ($10^{-6}$ μg C) is the carbon biomass of individual tintinnid, $V_i$ (μm$^3$) is the lorica volume.

21. L123-124 Something is wrong in this sentence. I think the word "test" should be deleted. Please check.

Response: We deleted the word "test" **in revised manuscript.**

22. L155-157 This is a lot of info which is not well-arranged. This info would be better presented in a table. Info later on in the text could also be included in such table.

Response: We accepted suggestions and added a table named "Table S1" accordingly **in revised Supplementary material.**

23. Figure 2 and caption (similar Figures 3 and 4): Please write in the caption which depths are shown, for example, … in panels from top to bottom 2m, etc.

Response: We accepted suggestions and revised accordingly **in lines 163–164, lines 203–204, and lines 206–207 in revised manuscript.**

Lines 163–164: Figure 2: Variations in environmental variables and ciliate abundance and biomass at discrete depth (2, 50, 100 and 200 m) in each temperature zone.

Lines 203–204: Figure 3: Variations in body-size spectra of ciliate normalized abundance at discrete depth (2, 50, 100 and 200 m) in each temperature zone.

Lines 206–207: Figure 4: Variations in body–size spectra of ciliate normalized biomass at discrete depth (2, 50, 100 and 200 m) in each temperature zone.

24. L177-179 "Generally, the slopes of the normalized abundance and biomass size spectra varied from -2.13 to -0.87 (average -1.60±0.33), and from -0.99 to -0.08 (average -0.53±0.25), respectively, with the former was much steeper than the latter (Figure 3)." This sentence is really hard to understand. Please split the sentence in two or three and explain exactly what you mean.

Response: We revised this sentence accordingly **in lines 186–188, lines 188–190, and lines 196–198 in revised manuscript.**

Lines 186–188: Generally, the slopes of the normalized abundance size spectra varied from -2.13 to -0.87 (average -1.60±0.33), and relevant biomass values varied from -0.99 to -0.08 (average -0.53±0.25), with the former slope line was much steeper than the latter.

Lines 188–190: Therein, ciliate abundance decreased from small (15 μm) to large size spectra (> 100 μm), with the slope line of the normalized abundance size spectra in both the NFZ (-2.13 to -1.93, average -2.01±0.09) and SFZ (-2.01 to -1.63, average -1.80±0.17) being steeper than in the other three regions at each depth.

Lines 196–198: Moreover, the slope lines of the normalized biomass size spectra in the SFZ (-0.99 to -0.77, average -0.86±0.10) were steeper than that in the SAZ (-0.74 to -0.43, average -0.62±0.13), NTZ (-0.63 to -0.44, average -0.53±0.09), TZ (-0.74 to -0.25, average -0.47±0.22) and NFZ (-0.37 to -0.08, average -0.21±0.12) (Figure 4).

25. L197-198 Please explain how the Margalef and Shannon indices are defined and what they are used for, as some readers are probably not familiar with that.

Response: We added the usages of both Margalef and Shannon indices accordingly **in lines 209–211 in revised manuscript.**

Lines 209–211: Tintinnid assemblages exhibited significant spatial heterogeneity in both species richness and diversity metrics (Margalef index–$d_{Ma}$ and Shannon index–$H_2'$ are quantitative measures of species richness in ecological communities) across five temperature zones.

26. L223 polar instead of bipolar

Response: We revised "bipolar" into "polar" accordingly **in lines 236–237 in revised manuscript.**

Lines 236–237: In addition, only the polar seas exhibited an increasing trend ($\Delta_1 \geq$ 0.01) in species richness–Chl $a$ correlation at each sampling layer (Figure S9).

27. L252-253 The sentence "Additionally, … and SFZ." Does not appear to fit here and should be deleted.

Response: We accepted suggestions and deleted the previous sentence **in revised manuscript.**

28. L254-255 "However, the current dataset remains geographically constrained, particularly lacking representation from Atlantic Ocean ecosystems …" This is an important contention which could be further discussed. Is there any work on ciliates done in the Atlantic which can be compared with the present study?

Response: Dear reviewer, to our knowledge, only Li et al. (2023) studied the tintinnid assemblage in the Atlantic Ocean as listed in the following text. However, this work on ciliates done in the Atlantic can not be compared with the present study due to the differences in sampling process. During the sampling process, Li et al. (2023) just got tintinnid samples from DCM to surface layers combined with 5-20 L of water gently filtered through a 10 mm mesh net. Unlike to our sampling process in the manuscript, this sampling method will cause greatly damage to aloricate ciliates (dominant group among pelagic ciliate globally). Therefore, we wrote that "However, the current dataset remains geographically constrained, particularly lacking representation from Atlantic Ocean ecosystems where ciliate communities may exhibit distinct adaptive strategies" **in lines 266–267 in revised manuscript.**.

Li, H., Tarran, G. A., Dall'Olmo, G., Rees, A. P., Denis, M., Wang, C., Gregori, G., Dong, Y., Zhao, Y., Zhang, W., and Xiao, T.: Organization of planktonic Tintinnina assemblages in the Atlantic Ocean. Front. Mar. Sci. 10, 1082495. doi:10.3389/fmars.2023.1082495, 2023

29. L264 change to: … the majority of integrative analyses …
Response: We accepted suggestions and revised accordingly **in lines 276–277 in revised manuscript.**
Lines 276–277: the majority of integrative analyses have primarily focused on biomass density within the size spectrum rather than on the abundance distribution across different trophic levels.

30. L268 polar instead of bipolar
Response: We revised "bipolar" into "polar" accordingly **in lines 279–280 in revised manuscript.**
Lines 279–280: our study revealed that the slopes of abundance size spectra in both the NFZ and SFZ were steeper in polar seas than other three regions latitudinally.

31. L288 polar instead of bipolar
Response: We revised "bipolar" into "polar" accordingly **in lines 299–300 in revised manuscript.**
Lines 299–300: Consistent with both observational and modeling studies, tintinnid biodiversity was highest in the tropical and subtropical seas, and was lowest in the polar seas.

32. L289-290 These are really many references for this contention. Maybe some of these can be discarded.

Response: We deleted five references in lines 299–302 in revised manuscript.

Lines 299–302: Consistent with both observational and modeling studies, tintinnid biodiversity was highest in the tropical and subtropical seas, and was lowest in the polar seas (Figure 5) (e.g., Sherr et al. 1997; Dolan et al. 2014, 2016; Righetti et al. 2019; Benedetti et al. 2021; Wang et al. 2020, 2024a; Li et al. 2016, 2018, 2022).

33. L292-293 change to: … more diversified phytoplankton is probably responsible for …

Response: We accepted suggestions and revised this sentence accordingly in lines 304–306 in revised manuscript.

Lines 304–306: more diversified phytoplankton in tropical zone (Tian et al., 2024) is probably responsible for subsequent higher tintinnid biodiversity compared to polar zones through endosymbiosis mechanism (Margulis and Sagan 2002; Clark et al. 2023).

34. L293 polar instead of bipolar

Response: We revised "bipolar" into "polar" accordingly in lines 304–306 in revised manuscript.

Lines 304–306: more diversified phytoplankton in tropical zone (Tian et al., 2024) is probably responsible for subsequent higher tintinnid biodiversity compared to polar zones through endosymbiosis mechanism (Margulis and Sagan 2002; Clark et al. 2023).

35. L292 "more diversified phytoplankton" Is that only existent in the tropical regions and not in the polar regions? That would need a reference

Response: We meant that more species richness occurred in the tropical regions compared to polar seas. In order to make this sentence more clear, we added several references accordingly in lines 303–306 in revised manuscript.

Lines 303–306: After a long-term genetic DNA exchange and evolution process driven by closely prey-predation interaction (Chen et al. 2012), more diversified phytoplankton in tropical zone (Tian et al., 2024) is probably responsible for subsequent higher tintinnid biodiversity compared to polar zones through endosymbiosis mechanism (Margulis and Sagan 2002; Clark et al. 2023).

36. L296-299 "Generally, large-scale hydrographic features, particularly oceanic gyres and distinct water masses, create biogeographic discontinuities that disrupt ecological connectivity despite physical ocean connectivity (Yang et al. 2020). These mesoscale structures establish unique ecoregions with characteristic environmental sensitivities (Longhurst 2007), …" This is important information. However, the interesting question would be if this also plays a role in the explanation of the structures in your observation? Please explain this in somewhat more detail.

Response: Our study indicated that tintinnid biodiversity was highest in tropical,

subtropical, temperate and polar seas might be due to physical barriers formed by oceanic gyres and distinct water masses. In order to make this sentence more clear, we revised this part accordingly **in lines 308–315 in revised manuscript.**

Lines 308–315: Generally, large-scale hydrographic features, particularly oceanic gyres and distinct water masses, create biogeographic discontinuities that disrupt ecological connectivity despite physical ocean connectivity (Yang et al. 2020). These mesoscale structures establish unique ecoregions with characteristic environmental sensitivities (Longhurst 2007), as evidenced by pronounced tintinnid community differentiation across the North Pacific Gyre, Subarctic Gyre, and Beaufort Gyre systems (Wang et al. 2020). Therein, our results revealed that tintinnid biodiversity was highest in the tropical (West Pacific and Indian Ocean) and temperate (North Pacific) seas, then followed by the Sub-Arctic (Bering Sea) and polar seas (Arctic Ocean and Southern Ocean around Antarctic) (Figure 5) were consistent with Wang et al. (2020), proved that plankton biogeography were deeply affected by oceanic gyres.

37. L310 "by affecting their thermal affinity within biogeochemical cycles" It is not clear to me what is meant here. Please explain and rephrase.

Response: For biota, temperature can promote their activity level through regulating intrinsic temperature-dependent metabolic processes. When the outer temperature were higher or lower than an organism's tolerance range, then the final ending for the organism was die. We aware that this sentence was not clear, and rephrased accordingly **in lines 321–324 in revised manuscript.**

Lines 321–324: Conventionally, temperature can impact plankton biodiversity through regulating intrinsic temperature-dependent metabolic processes, which further determined that which kind of species can live in such a specific temperature environment (Archibald et al. 2022; Lukić et al. 2022; Weisse 2024).

38. L314-316 "Our study, along with others, indicates that ciliate inhabiting higher salinity environments in both the TZ and NTZ (Figure S8) compared to bipolar regions might be a reflection of their 315 higher osmotic pressure affinity." This is an interesting conclusion. However, is there any indication that these relatively small salinity difference do play such a role? Are there laboratory experiments available to show that?

Response: Dear editor, targeted this problem, we just found that ciliate lived in a higher salinity environments in both the TZ and NTZ than polar seas (Figure S8), thus we made a speculation for this phenomenon. There were no relative laboratory experiments available to show the above-mentioned phenomenon.

39. L315 polar instead of bipolar

Response: We revised "bipolar" into "polar" accordingly **in lines 330–332 in revised manuscript.**

Lines 330–332: Our study, along with others, indicates that ciliate inhabiting higher salinity environments in both the TZ and NTZ (Figure S8) compared to polar regions might be a reflection of their higher osmotic pressure affinity.

40. L317-319 "Furthermore, the Chl a functionally serves as a critical ecological mediator in marine food webs, influencing ecosystem stability through both quantitative (abundance) and qualitative (polyunsaturated fatty acid composition) pathways via the fundamental prey-predator interplay …" I do not understand what the authors are saying here. All phytoplankton have chlorophyll, so why would chl be the mediator? And what has chl to do with the polyunsaturated fatty acid composition? Please explain.

Response: We revised this sentence accordingly in order to state the ecological function of Chl *a* accordingly **in lines 333–335 in revised manuscript.**

Lines 333–335: Furthermore, the Chl *a* functionally serves as the food resource in marine food webs, influencing ecosystem stability through both quantitative (abundance) and qualitative (nutrient composition) pathways via the fundamental prey-predator interplay (Šolić et al. 2010; Våge and Thingstad 2015; Holm et al. 2022).

41. L319-320 "Consequently, Chl a modulated the energy flow of the entire marine ecosystem (Li et al. 2024)." This is trivial, because all energy in the system in the end originates from phytoplankton and its chl.

Response: We deleted this sentence **in revised manuscript.**

42. L326-327 "in structuring global microzooplankton communities." Your results cannot just be extrapolated to all microzooplankton. You can only contend this for the ciliates in your study.

Response: We accepted suggestions and revised accordingly **in lines 338–342 in revised manuscript.**

Lines 338–342: while observed trait plasticity in ciliate communities (Yu et al. 2022) further supports the predominance of bottom-up control mechanisms (resource availability, prey quality) (Lu and Weisse 2022; Wang et al. 2023c, 2024c) over top-down regulation (predation pressure from microcrustaceans) (Power 1992; Calbet et al., 2001; Worm and Myers, 2003) in structuring global pelagic ciliate communities.

43. L327-328 "This trophic cascade pattern underscores the fundamental role of primary production dynamics in governing ciliate population ecology across marine ecosystems." This sounds trivial. Maybe you can be more specific about what you mean here.

Response: We deleted this sentence accordingly **in revised manuscript.**

44. L330 delete industrial-induced

Response: We deleted "industrial-induced" accordingly **in revised manuscript.**

45. L340 delete full sentence as this was already stated in the previous paragraph

Response: We deleted this full sentence (previous L340) **in revised manuscript.**

46. L342 delete Similarly

Response: We deleted "Similarly" **in revised manuscript.**

47. Discussion 4.4 This is an interesting discussion. However, I would expect that the direct results by the authors would play a bigger role in this discussion. I encourage the authors to expand the discussion to some extent to include concrete ciliate results

Response: We added a sentence and revised this part accordingly **in lines 362–365 in revised manuscript.**

Lines 362–365: Moreover, combined with our results that only the NFZ and SAZ exhibited an increasing trend ($\Delta_I \geq 0.03$) in abundance–emperature correlation at surface layers compared with other three zones (Figure S9), we predict that the pelagic surface‐dweller ciliates in both the sub-Arctic and Arctic seas will benefit from the future global warming..

48. L358 polar instead of bipolar

Response: We revised "bipolar" into "polar" accordingly **in lines 373–374 in revised manuscript.**

Lines 373–374: Additionally, tintinnid biodiversity was highest in tropical and subtropical seas and lowest in polar seas.

49. All references: Please place a comma after the author's initial. For example, the first reference will then read: Amargant-Arumí M., Müller, O., Bodur, Y., Ntinou, I., Vonnahme, T., Assmy, P., Kohlbach, D., Chierici, M., Jones, E., Olsen, L., Tsagaraki, T., Reigstad, M., Bratbak, G., and Gradinger, R.:

Response: We added a comma after the author's initial throughout all references **in revised manuscript.**

---

## Author Response (AR3)

**Dear Editor**,

We finished the revision of the manuscript according to the questions and advices of the reviewer. The following are the details of our responses (in blue color) to questions and advices of the reviewer.

The work of reviewers help improve the quality of the manuscript. We thank the thoughtful advice of the reviewer and hope the revision successfully answered the questions.

Best wishes

**Wuchang Zhang**
=============================================================
Public justification (visible to the public if the article is accepted and published):

Dear authors,

Your manuscript can now be accepted for publication in Ocean Science. Please take into account the final technical comments.

Additional private note (visible to authors and reviewers only):

Please make to final modifications before sending your final version to the production department. I am using the numbers that you use in your author's response:

1.  Your number 5: The slope line that you changed to is better. However, I think the concept of slope or slope line is not clear to the reader. Please insert in the introduction the text that you give in the response: "(tendency for evaluating the decreasing trend from small to large size spectrum)"

Response: We revised this part accordingly **in lines 114–115 in revised manuscript.**

Lines 114–115: In addition, the slope or slope line means tendency for evaluating the decreasing trend from small to large size spectrum.

2.  Your number 9 (lines 31-33): "generalised" seems to be the wrong term here. I think it would be better served with "taken as a guideline" or similar: "… ciliate trait structure among five temperature zones and can be taken as a guideline for assessing the potential effects of climate change on pelagic ciliates in future marine realm."

Response: We accepted suggestions and revised accordingly **in lines 31–33 in revised manuscript.**

Lines 31–33: In conclusion, these results underscore the unprecedented divergences in ciliate trait structure among five temperature zones and can be taken as a guideline for assessing the potential effects of climate change on pelagic ciliates in future marine realm.

3.  Your number 11: I think Singh et al. is still not the correct reference for "Over recent decades, anthropogenic $CO_2$ emissions have led to increased atmospheric concentrations and greater global radiative forcing" The safe reference here is the

IPCC.

Response: We revised into "IPCC 2023" accordingly **in lines 47–48 in revised manuscript.**

Lines 47–48: Over recent decades, anthropogenic $CO_2$ emissions have led to increased atmospheric concentrations and greater global radiative forcing (IPCC 2023), triggering diverse ecological feedbacks worldwide…

4.    Your number 18: It is unusual that the precision of measurements is unknown. I urge you to give some kind of precision, because without a precision/accuracy one cannot know the reliability of the data. The fact that you use a Seabird CTD may give you a solution at hand, as the Seabird will give in the specifications the possible precision that can be reached. Please use that in the manuscript.

Response: We added the precision of measurements of a Seabird CTD for environmental factors (depth, temperature, salinity and Chl *a*) during each cruise accordingly **in lines 115–122 in revised manuscript.**

Lines 115–122: Simultaneously, environmental factors of sampling depth (a quartz pressure sensor to detect hydrostatic pressure, converted to depth via the formula: Depth = Pressure/[$\rho \times g$], where $\rho$ is water density and $g$ is gravitational acceleration) (van Haren et al., 2021), temperature (a thermistor, SBE-3 Plus, resolution is 0.0001°C), salinity (derived from measured electrical conductivity [SBE-4C sensor] and temperature data, computed using the Practical Salinity Scale algorithm) and chlorophyll *a* in vivo fluorescence (Chl *a*, a fluorometer [SeaPoint] excites chlorophyll pigments with blue light and measures emitted red light intensity as a proxy for Chl *a* concentration) were recorded by a multi-sensor profiler (CTD–SeaBird                                      SBE                                      911, https://www.seabird.com/product.detail-cms.block.jsa?id=60761421595) during each cruise.

5.    Your number 40: Your write in lines 333-335: "Furthermore, the Chl a functionally serves as the food resource in marine food webs …" Chl a is of course not a food resource, not even functionally. Phytoplankton may be the food, but not chl a as such. This has to be changed.

Response: We revised this sentence accordingly **in lines 339–341 in revised manuscript.**

Lines 339–341: Furthermore, the Chl *a* is roughly represent of phytoplankton at specific sampling layer, which further influencing marine ecosystem stability through both quantitative (abundance) and qualitative (nutrient composition) pathways via the fundamental prey-predator interplay (Šolić et al. 2010; Våge and Thingstad 2015; Holm et al. 2022).